# What Governs the Quality-Aware Generalization and Representation Capacity in No-Reference Image Quality Assessment Models?

## Abstract

Due to the high annotation costs and relatively small size of existing Image Quality Assessment (IQA) datasets, attaining both consistent generalization and quality representation capacity remains a significant challenge for prevalent deep learning (DL)-based Blind IQA (BIQA) methods. Although effective representation learning for distortion is deemed crucial for the generalization of the BIQA method, the theoretical underpinnings for this belief remain elusive. Therefore, in this study, we innovatively explore the theoretical quality-aware generalization bounds and representation capacity of DL-based IQA models, as well as the relationship between their respective determinants. For the generalization bound, under the assumption that training and test distributions are identical, we derive the fine-grained and coarse-grained upper bounds for BIQA generalization errors using covering number and VC dimension, respectively. These two theoretical results are presented in Theorem 1 and Theorem 2, revealing the role of low-level features in generalization. Under distribution shifts, we propose a tighter generalization bound to investigate the impact of distributional differences between training and test sets on BIQA generalization in Theorem 3 using Intrinsic Dimension, which can further confirm the generalization role of low-level features. For quality representation capacity, in Theorem 4, we quantify the representation capacity for BIQA models based on PAC-Bayes. The theoretical result demonstrates that learning higher-level quality features can enhance the quality representation capacity. These theorems offer theoretical support for enhanced performances in existing BIQA methods. Interestingly, our theoretical findings reveal an inherent tension between robust generalization and strong representation capacity in BIQA, which motivates effective strategies to lower empirical errors without undermining generalization. Extensive experiments confirm the reliability and practical value of our theorems.

## 1 Introduction

Image quality assessment (IQA) plays a crucial role in optimizing visual experiences across different domains, including image denoising (Tian et al., 2020), restoration (Cui et al., 2023), and generation (Elasri et al., 2022). Its primary objective is to develop algorithms that can predict image quality scores consistent with subjective human ratings (i.e., Mean Opinion Scores, MOS). Based on the availability of reference information, existing IQA methods can be classified into Full-Reference IQA (FR-IQA), Reduced-Reference IQA (RR-IQA), and Blind IQA (BIQA) (Zhai & Min, 2020). Among these, BIQA exhibits broader applicability due to its independence from the reference image (Simeng et al., 2023; Feng et al., 2021). Traditional BIQA methods aim to evaluate the perceptual quality by manually extracting or selecting valid statistical features from distorted images (Jiang et al., 2017; Liu et al., 2020; Zhou et al., 2017). These methods have shown promising results in evaluating images with synthesized distortions, while they perform poorly in authentically distorted scenarios. Consequently, many deep learning (DL)-based BIQA methods have been introduced to handle authentic distortions by leveraging powerful visual features (Ke et al., 2021; Madhusudana et al., 2022; Simeng et al., 2023; Talebi & Milanfar, 2018). However, the training process of these methods requires substantial data to avoid overfitting, while no large database of authentically distorted images is currently available due to the costly annotation process, leading to limited generalization (Prabhakaran & Swamy, 2023; Yue et al., 2022).

To address this issue, one of the most intuitive and effective ideas is to explore more effective network architectures and training paradigms for IQA tasks (Liu et al., 2017; Lin et al., 2020; Ma et al., 2017; Zhou et al., 2022). These studies have revealed that the enhanced generalization and quality feature representation can be attained through either complex structural designs or knowledge injection, but at the expense of greater training overhead. Therefore, the more economical design schemes are worthy of investigation, which aim to attain a significant boost in both generalization and quality representation capacity by only trading off a marginal efficiency loss without extra training modules or data annotations. Although this insight has inspired the core idea of hierarchical feature fusion in numerous BIQA methods (such as MUSIQ (Ke et al., 2021), Hyper-IQA (Su et al., 2020), and Stair-IQA (Sun et al., 2022), etc), the theoretical foundations remain underexplored, leading to poor interpretability of the success of these methods. Moreover, existing theoretical results about neural networks mainly target classification tasks and fail to offer convincing support in the IQA domain. Hence, in this work, we theoretically investigate the generalization ability and quality representation capacity of the BIQA framework in a fully-supervised regressive setting.

For a systematic theoretical analysis of *what governs the generalization bounds and representation capacity in BIQA*, we base our investigation on an unembellished CNN-based BIQA model, setting aside the complex structural designs or additional knowledge injection. The core theoretical contributions comprise five novel theorems establishing fundamental relationships between the BIQA model architectures and two key aspects: the generalization bound and the quality representation capability.

- **Generalization Upper Bound.** (1) Under distribution invariance in training and test sets, we firstly use *Covering number* (Bartlett, 1998) to derive a coarse-grained upper bound of the generalization error for a CNN-based BIQA model in Theorem 1. Then, to obtain a fine-grained generalization bound, we provide Theorem 2 from the perspective of *VC dimension* (Sepliarskaia et al., 2024a). These two theoretical results both imply that as the level of quality features increases, the generalization ability tends to drop, highlighting the value of low-level features for quality perception. (2) Under distribution shifts, we propose a tighter generalization bound based on *Rademacher complexity* (Kakade et al., 2008) in Theorem 3, which further reveals how distribution differences between training and test sets affect generalization. In addition, the *Intrinsic Dimension* can further confirm the generalization role of low-level features.
- **Quality Representation Capacity.** We show that BIQA networks tend to possess smaller empirical errors and higher divergence between the posterior distribution and the prior distribution of BIQA models in the hypothesis space. Inspired by *Probably Approximately Correct Bayesian (PAC-Bayes)* (McAllester, 1999), the divergence between posterior and prior model distributions can measure the representation capacity; hence, we derive the divergence to obtain the theoretical quality-aware representation capacity of BIQA models based on *PAC-Bayes* in Theorem 4, which indicates that focusing on higher-level image features can boost quality representation capability.

These theoretical findings help explain why many BIQA methods, despite different structural designs, manage to achieve good generalization and representation capability. They also unveil a fundamental conflict between generalization (favoring lower-level features) and representation capacity (favoring higher-level features). This conflict inspires further work on designing BIQA strategies that reduce empirical errors without undermining generalization. On this basis, we offer some exploratory suggestions for BIQA training to balance this trade-off, and cross-dataset experiments show that these suggestions are effective in practice. In summary, the principal contributions of this work are:

- We provide a theoretical analysis about the upper bounds of the generalization and quality representation capabilities for CNN-based BIQA models, identifying their primary determining factor: the network attributes governing the hierarchical levels of the extracted quality features. This work marks the first explicit theoretical treatment of IQA generalization and representation capacity.
- Under the conditions of distribution invariance or shift between training and test datasets, we respectively provide the multi-granularity generalization bounds for CNN-based BIQA networks, with rigorous proofs emphasizing the crucial role of low-level features in generalization.
- Through the analysis based on PAC-Bayes, we provide the upper bound of quality representation capacity, and prove that the BIQA networks focused on learning higher-level features tend to exhibit lower empirical errors and stronger representation capacities in quality perception, which validates the importance of high-level features.
- We uncover the fundamental conflict between the generalization and representation capacities of CNN-based BIQA models. Correspondingly, the proposed Theorems can offer theoretically

valid suggestions for BIQA training. Experimental results demonstrate the effectiveness of these suggestions, reflecting the reliability and practical value of our theoretical findings.

We provide Related Works in Appendix B, including BIQA and generalization bound in deep learning.

## 2 PRELIMINARIES

### 2.1 UNEMBELLISHED CNN-BASED BIQA MODEL

Without loss of generality, we consider a standard CNN-based BIQA network with $L$ layers following the architecture in (Sun et al., 2016). This includes $L - 1$ hidden layers for quality perception feature extraction and an output layer for MOS prediction. Suppose each $l$-th layer ($l = 0, \cdots, L$) has $m_l$ units, with $m_L = 1$ for regression training and reference. To control overfitting, it is standard to constrain the sum of the weight magnitudes for each unit by a factor $A$. We denote the function space of such CNN-based IQA networks with depth $L$ by $\mathcal{F}_L$. Specifically, for convenience of presentation, we first denote the input layer as follows for the input image $x$, formalized as the function class $\mathcal{F}_0$:

$$\mathcal{F}_0(x) \leftarrow x_{p_1, p_2, p_3} \quad (1 \le p_1 \le H, 1 \le p_2 \le W, 1 \le p_3 \le 3), \tag{1}$$

where $x_{p_1, p_2, p_3}$ is the pixel value in image $x$ with the height, width, and channel index $p_1, p_2, p_3$.

Then, the hidden layers (formalized as the function classes $\mathcal{F}_l, l = 1, \cdots, L - 1$) are computed by two steps: (1) Linear transformation and activation: perform a linear combination of the outputs from the previous layer, followed by an activation function $\phi$ (e.g., Sigmoid, ReLU). (2) Pooling operation: apply pooling (e.g., max pooling or average pooling) to the activated results, denoted by the function $\varphi_{p_l}$, where $p_l$ is the size of the pooling region in the $l$-th layer. The specific form of $\mathcal{F}_l$ is given by:

$$\mathcal{F}_l(x) \leftarrow \varphi_{p_l} \left( \phi \left( \sum_{j=1}^{m_{l-1}} w_j f_j(x) \right) \right), \quad \text{where} \quad f_j \in \mathcal{F}_{l-1}, \quad \sum_{j=1}^{m_{l-1}} |w_j| \le A. \tag{2}$$

In Eq. (2), $w_i$ is the weight parameter in the BIQA network. If the hidden layer is a fully connected layer, the pooling function reduces to the identity mapping (i.e., $\varphi_{p_l}(t) = t$ with $p_l = 1$); if it is a convolutional layer, the weights $w_j$ exhibit sparsity, and $m_l$ is determined by $m_{l-1}$ along with the number of the convolution kernels with domain size $k_l$ in the hidden layer $\mathcal{F}_l$. Note that the frequently employed activation functions are typically 1-Lipschitz, such as the sigmoid function, the hyperbolic tangent function, and the rectifier function.

Finally, the goal of the output layer (formalized as the function class $\mathcal{F}_L$) is to predict the MOS score for input image $x$ based on the extracted quality features, requiring only a linear transformation:

$$\mathcal{F}_L(x) \leftarrow \sum_{j=1}^{m_{L-1}} w_j f_j(x), \quad \text{where} \quad f_j \in \mathcal{F}_{L-1}, \quad \sum_{j=1}^{m_{L-1}} |w_j| \le A. \tag{3}$$

During training, the back-propagation method is typically utilized to minimize the empirical loss on the training set, where the weight parameters undergo updates through stochastic gradient descent.

### 2.2 EXPECTED AND EMPIRICAL ERRORS OF BIQA MODELS

In the fully-supervised BIQA tasks, we assume $\mathcal{X} \subseteq \mathbb{R}^{H \times W \times 3}$ as the input space of distorted images, $\mathcal{Y} \subseteq [a, b]$ as the output space of MOS labels. $P$ is the joint distribution over $\mathcal{X} \times \mathcal{Y}$. We denote $S = \{(x_1, y_1), \cdots, (x_n, y_n)\}$ as the training dataset, each sample of which is i.i.d. sampled from $\mathcal{X} \times \mathcal{Y}$ based on the distribution $P$. The objective is to train an effective MOS prediction model on $S$, i.e., $f \in \mathcal{F} : \mathcal{X} \to \mathbb{R}$ with the regression loss such as $L_1$ loss. Therefore, the empirical error $\text{err}_S(f)$ and expected error $\text{err}_P(f)$ of the BIQA model $f$ on the test set can be measured by:

$$\text{err}_S(f) = \frac{1}{n} \sum_{i=1}^{n} |f(x_i) - y_i| \quad \text{and} \quad \text{err}_P(f) = \mathbb{E}_{(x,y) \sim P} |f(x) - y|. \tag{4}$$

The empirical error $\text{err}_S(f)$ in Eq. (4) evaluated on the training set $S$ is the training error. The expected error $\text{err}_P(f)$ on the test set is termed the test error, which serves as a proxy for assessing the prediction accuracy due to the unknown underlying data distribution $P$. To theoretically expose the effects of different architecture parameters of the BIQA model on the generalization performance, we first prove that the setting of $L_1$ loss in BIQA tasks satisfies the Lipschitz condition (Valentine, 1945). It is a necessary prerequisite for establishing the BIQA generalization bound subsequently.

**Lemma 1.** *The $L_1$ loss function satisfies the Lipschitz condition with Lipschitz constant $L_\ell = 1$.*

*Proof.* Please refer to the Appendix C. □

Building upon Lemma 1, we can derive the upper bound of the expected error, which provides an intuitive reflection of the generalization upper bound (i.e. $\mathrm{err}_P(f) - \mathrm{err}_S(f)$) and generalization capability for BIQA models trained with the $L_1$ loss. The remainder of this paper is organized as follows: Sections 3 and 4 establish theoretical upper bounds for the generalization error and representation capability of CNN-based BIQA models under different conditions. Sections 5 and 6 provide the in-depth discussion and experimental validation of our theoretical results and findings.

# 3 GENERALIZATION ERROR BOUND OF BIQA MODELS

In this section, we systematically analyze the generalization bounds for BIQA models under two distinct scenarios: distribution invariance and distribution shift between training and test sets.

## 3.1 GENERALIZATION BOUNDS UNDER DISTRIBUTION INVARIANCE

For a systematic analysis of generalization under distribution invariance, we establish both fine-grained and coarse-grained upper bounds for the generalization of BIQA models.

**For the coarse-grained generalization bound**, we establish it based on the *Covering number* theory (Zhou, 2002; Zhang, 2002). *Covering number* is a fundamental tool in generalization theory by quantifying the minimum cardinality of an $\epsilon$-*covering* of function class $\mathcal{F}$ under the supremum norm, representing the smallest number of functions required to approximate any element in $\mathcal{F}$ within precision $\epsilon$. Following (Bartlett, 1998) and (Shen, 2024), we first introduce the relevant definitions on the concept of *Covering number*.

**Definition 1** ($\epsilon$–cover). *Let $(X, \rho)$ be a metric space and $H \subseteq X$ a subset. A subset $G \subseteq X$ is called an $\epsilon$–cover of $H$ with respect to $\rho$ if for every $h \in H$ there exists $g \in G$ satisfying $\rho(h, g) \leq \epsilon$. The size of the smallest $\epsilon$–cover of $H$ is denoted by $\mathcal{N}_\rho(H, \epsilon)$.*

**Definition 2** (Covering number). *For a domain $Z$, define a metric $\rho_{\max}(f, g) = \sup_{z \in Z} |f(z) - g(z)|$ on pairs of functions $f, g : Z \to \mathbb{R}$. $\mathcal{N}_{\rho_{\max}}(G, \varepsilon)$ is the covering number of $G$ with respect to $\rho_{\max}$.*

**Definition 3** (Covering number with supremum norm). *Let $\mathcal{F} = \{f : \mathcal{X} \to \mathbb{R}\}$ be a class of functions. The supremum norm of $f \in \mathcal{F}$ is defined as $\|f\|_\infty := \sup_{x \in \mathcal{X}} |f(x)|$. Then, for a given $\epsilon > 0$, we define the covering number of $\mathcal{F}$ with radius $\epsilon$ under the norm $\|\cdot\|_\infty$ as the least cardinality of a subset $\mathcal{G} \subseteq \mathcal{F}$ satisfying $\sup_{f \in \mathcal{F}} \min_{g \in \mathcal{G}} \|f - g\|_\infty \leq \epsilon$, which is denoted by $\mathcal{N}(\mathcal{F}, \epsilon, \|\cdot\|_\infty)$.*

Based on the above Definitions 1-3, we bound the covering number of the CNN-based BIQA model in Eqs. (1-3) to derive its generalization upper bound. For simplicity, here, we assume that each convolutional layer is parameterized by its convolutional kernels, where the $l$-th layer has $d_l$ filters and each filter has $k_l$ parameters[1], and that all the parameters are bounded by $a$ in a suitable norm.

**Theorem 1.** *Assume that the CNN-based BIQA model has $L$ layers, the $l$-th layer has $d_l$ filters with kernel size $k_l$, all the parameters are bounded by $a$, the metric $\rho_{\max}$ is $\|\cdot\|_\infty$ metric, and $n$ is the number of training samples. Then the covering number bound for the hypothesis class $\mathcal{F}_L$ is:*

$$\log \mathcal{N}(\epsilon, \mathcal{F}_L, \|\cdot\|_\infty) \leq \sum_{l=1}^{L} k_l d_l \log\left(\frac{a}{\epsilon}\right). \tag{5}$$

*Consequently, with probability at least $1 - \delta$, every model $f \in \mathcal{F}_L$ satisfies the generalization bound:*

$$\mathrm{err}_P(f) \leq \mathrm{err}_S(f) + O\left(\frac{1}{n}\sqrt{\sum_{l=1}^{L} k_l d_l \log(a/\epsilon) + \log(1/\delta)}\right). \tag{6}$$

Based on Theorem 1, we can obtain a tighter generalization bound under a stricter assumption.

**Corollary 1** (Tighter Generalization Bound). *Under the same assumptions as in Theorem 1, suppose further that the total number of learnable parameters across all $L$ layers scales at most linearly with the depth, i.e., $\sum_{l=1}^{L} k_l m_l = O(L)$. Then the generalization bound can be tightened to:*

$$\mathrm{err}_P(f) \leq \mathrm{err}_S(f) + O\left(\frac{1}{n}\sqrt{L^2 \log(a/\epsilon) + \log(1/\delta)}\right), \tag{7}$$

*Proof.* Refer to Appendix D.1 and Appendix D.2 for the proofs of Theorem 1 and Corollary 1. □

---

[1]This typically comes from the spatial kernel size $k_l$, we absorb the number of parameters into $k_l$.

From the above theoretical results, we can make the following key observations and conclusions: **(1)** The increasing depth and kernel size lead to a looser generalization bound, indicating that relying solely on high-level quality perception features can have a negative impact on the generalization ability of BIQA models. This highlights the importance of low-level image features for the generalization of IQA. **(2)** As the sample size $n$ increases, the generalization bound will decrease, which aligns with the common understanding of neural network training. This also theoretically confirms that insufficient training data is a major reason behind the poor generalization ability of existing BIQA models. **(3)** The tighter weight parameter boundary may promote better generalization. This motivates us to apply a regularization penalty on the model parameters during the training process.

**For the fine-grained generalization bound**, through the Vapnik–Chervonenkis theory (Vapnik & Chervonenkis, 1971; Vapnik, 1982; Pollard, 1984; Haussler, 1992), we provide the fine-grained bound based on *VC dimension*, the definition of which is as follows.

**Definition 4** (Growth function, VC dimension, shattering (Sepliarskaia et al., 2024a)). *Let $\mathcal{H}$ be a class of functions mapping a domain $\mathcal{F}$ to $\{-1, 1\}$ (the* hypothesis class*). For any non-negative integer $m$, the* growth function *of $\mathcal{H}$ is defined as $\Pi_{\mathcal{H}}(m) := \max_{f_1,\ldots,f_m \in \mathcal{F}} \left| \left\{ (h(f_1), \ldots, h(f_m)) : h \in \mathcal{H} \right\} \right|$. If $\mathcal{H}$ can realise all $2^m$ possible dichotomies on a set of $m$ inputs, we say that $\mathcal{H}$ shatters that set. Formally, if $\left| \left\{ (h(f_1), \ldots, h(f_m)) : h \in \mathcal{H} \right\} \right| = 2^m$, then $\mathcal{H}$ shatters $\{f_1, \ldots, f_m\}$. The Vapnik–Chervonenkis dimension of $\mathcal{H}$, denoted $\mathrm{VC}(\mathcal{H})$, is the largest $m$ such that $\Pi_{\mathcal{H}}(m) = 2^m$; if no such largest $m$ exists, we set $\mathrm{VC}(\mathcal{H}) = \infty$.*

From Definition 4, the classical VC dimension applies only to Boolean-valued function classes. For real-valued hypothesis classes—such as neural networks, we follow (Bartlett et al., 2019) and adopt the *Pseudodimension*, which preserves the same uniform-convergence properties (Pollard, 1990; Anthony & Bartlett, 1999).

**Definition 5** (Pseudodimension (Sepliarskaia et al., 2024a)). *For a real-valued function class $\mathcal{H}$, define $\mathrm{sign}(\mathcal{H}) := \left\{ \mathrm{sign}(H - b) \mid H \in \mathcal{H}, b \in \mathbb{R} \right\}$, where $\mathrm{sign}(x) = 1$ for $x > 0$ and $-1$ otherwise. The pseudodimension of $\mathcal{H}$ is then $\mathrm{VC}(\mathcal{H}) := \mathrm{VC}\big(\mathrm{sign}(\mathcal{H})\big)$. $\Pi_{\mathcal{H}}$ is the growth function of $\mathrm{sign}(\mathcal{H})$.*

For real-valued functions whose outputs are clipped to $[-1, 1]$, the 0–1 loss can be upper-bounded by the $L_1$ loss. Thus, via the classic Vapnik–Chervonenkis inequality (Theorem 12.5 in (Anthony & Bartlett, 1999)), we have:

**Lemma 2.** *Let $\mathcal{F}_L$ be a CNN-based BIQA model class with VC dimension $d_{\mathrm{VC}}$ and $e$ be the base of natural logarithm. For any $\delta \in (0, 1)$, with probability at least $1 - \delta$ over an i.i.d. sample of size $n$,*

$$\mathrm{err}_P(f) \leq \mathrm{err}_S(f) + \sqrt{\frac{8}{n}\left(d_{\mathrm{VC}}\big(\log\frac{2en}{d_{\mathrm{VC}}}\big) + \log\frac{4}{\delta}\right)}, \quad \forall f \in \mathcal{F}_L. \tag{8}$$

*Proof.* Please refer to the Appendix E. $\square$

Based on Lemma 2, we can obtain the following fine-grained generalization bound by giving a tight estimate of $d_{\mathrm{VC}}$ for the CNN-based BIQA models with depth $L$ and width $m_i$ in the $i$-th layer:

**Theorem 2.** *Consider the CNN-based BIQA class $\mathcal{F}_L$ defined in Eqs. (1-3) is the Growth function $\mathcal{H}(k, m_0, \ldots, m_L, r)$, where $L$ is the number of layers, $m_i$ is the width of the network in the $i$-th layer, $k$ is the kernel size (the number of parameters associated with the local receptive field in each unit), and $r$ is a bound on the output range determined by a scaling parameter. For any $\delta \in (0, 1)$, with probability at least $1 - \delta$ over an i.i.d. sample of size $n$, any function $f \in \mathcal{F}_L$ satisfies:*

$$\mathrm{err}_P(f) \leq \mathrm{err}_S(f) + O\left(\frac{1}{\sqrt{n}}\left(\min\left\{\sqrt{kL^2 \log(rL) \log\big(\frac{n}{kL^2 \log(rL)}\big)}, \sqrt{8\big(kL^2 \log\big(\frac{nr}{k}\big)\big)}\right\} + \sqrt{\log\big(\frac{1}{\delta}\big)}\right)\right) \tag{9}$$

*Proof.* Please refer to the Appendix F. $\square$

Theorem 2 establishes a more fine-grained generalization bound for CNN-based BIQA models, which aligns with the conclusions of Theorem 1. Specifically, it demonstrates that the dataset size $n$, kernel

size $k$, and network depth $L$ collectively govern the generalization bound, highlighting the role of low-level quality-aware features. Additionally, two novel insights emerge: **(1)** the scaling parameter $r$ also influences the generalization bound, and the logarithmic term $\log(rL)$ captures the compounded multiplicative effect of expanding output ranges and increasing network depth on the bound. **(2)** the theoretical generalization bound holds under the condition that the number of training samples is sufficiently large. Otherwise, the logarithmic term becomes negative and invalidates the bound.

### 3.2 GENERALIZATION BOUNDS UNDER DISTRIBUTION SHIFT

In Theorems 1 and 2, the generalization bound is not tight enough since it exhibits near-linear yet supra-linear growth with depth $L$. In addition, the theoretical results in Eqs. (6,7,9) ignore the effect of distribution difference on the generalization of BIQA models. Nevertheless, the effect of the distribution shift from training set to test set on generalization performance is significant, leading to a series of domain adaptation efforts on the IQA domain. Thus, it is meaningful to propose a tighter generalization boundary related to the distribution difference between training and test sets in BIQA.

Based on Lemma 1, the $L_1$ loss used in BIQA tasks is 1-Lipschitz, hence we can derive the generalization error bound with regard to *Rademacher complexity* (Kakade et al., 2008).

**Lemma 3** ((Bartlett & Mendelson, 2002)). *Assume the loss $\ell$ is Lipschitz (with respect to its first argument) with Lipschitz constant $L_\ell$ and that $\ell$ is bounded by $c$. For any $\delta > 0$ and with probability at least $1 - \delta$ simultaneously for all $f \in \mathcal{F}$, we have the upper bound of the expected error:*

$$\text{err}_P(f) \leq \text{err}_S(f) + 2L_\ell \mathcal{R}_n(\mathcal{F}) + c\sqrt{\frac{\log(1/\delta)}{2n}} \tag{10}$$

*where $n$ is the sample size. $\mathcal{R}_n(\mathcal{F})$ is the Rademacher complexity (Kakade et al., 2008) of the function class $\mathcal{F}$, details of which are shown in Definition 8 in the Appendix H.*

To obtain the generalization bound for CNN-based BIQA models, let $\mathcal{F}$ in Lemma 3 be the function class of $\mathcal{F}_L$ in Eqs. (1-3). Since the convolutional and fully-connected layers can be uniformly represented by Eq. (2), and because the functionality of CNN can also be implemented by MLP (Tolstikhin et al., 2021), for analytical convenience, we consider BIQA network $f$ as a unified MLP, including all sub-MLPs in different levels of quality perception representations. This transformation is mathematically equivalent, as all parameters and activation functions in this unified MLP can be derived from the original model without extra computation (Wu et al., 2024c). From Definition 8 in Appendix H and (Golowich et al., 2018), we have:

**Lemma 4** ((Golowich et al., 2018)). *Let $n$ be the number of image samples in the training set, $W_j$ the parameter matrix in the $j$-th layer, $M_F(j)$ the upper-bound of $\|W_j\|_F$, and $L$ the number of layers. For the BIQA model class $\mathcal{F}$, the Rademacher complexity is bounded by:*

$$R_n(\mathcal{F}) \leq \frac{L\log 2}{n\lambda} + \frac{1}{n\lambda}\log\left(\cdot\mathbb{E}_\epsilon \exp\left(M\lambda Q\right)\right), \quad M = \prod_{j=1}^{L} M_F(j), Q = \left\|\sum_{i=1}^{n} \epsilon_i x_i\right\|, \tag{11}$$

*where $x_i$ denotes the $i$-th instance, $\epsilon_i$ is a Rademacher variable, $\lambda$ is a random variable.*

To more accurately investigate the role of low-level quality features in the generalization of BIQA models, in the sample space $S = \{x_i, y_i\}$, we consider that there exists a low-dimensional structure in the high-dimensional space. Thus, we first adopt the concept of *Intrinsic dimension* (Tropp, 2015):

**Definition 6** (Intrinsic dimension (Tropp, 2015)). *The intrinsic dimension of a distribution $(\mu, \Sigma)$ is the ratio $1 \leq h(\Sigma) := \text{tr}(\Sigma)/\|\Sigma\|_2 \leq d$, where $\|\Sigma\|_2$ is the spectral norm (largest eigenvalue) of $\Sigma$.*

Based on Definition 6, we can obtain the measured dimension of the low-level quality features.

**Lemma 5** (Measured dimension of low-level features). *Let $\mathcal{F}_1(x) \in \mathbb{R}^d$ be a random feature whose support is contained in an $h$-dimensional subspace of $\mathbb{R}^d$. Let $\Sigma$ be its covariance matrix, and suppose $\Sigma$ has rank $h$ with non-zero eigenvalues $\lambda_1, \lambda_2, \ldots, \lambda_h$. If these $r$ eigenvalues are all equal (that is, $\lambda_1 = \lambda_2 = \cdots = \lambda_h > 0$), then the dimension measure $h(\Sigma) = \text{tr}(\Sigma)/\|\Sigma\|_2$ coincides with $h$. In other words, when the variance is equally distributed among the active directions, the measured dimension of the low-level quality features equals the intrinsic dimension $h$.*

*Proof.* Please refer to the Appendix G. $\square$

Based on the above Lemmas, we are now ready to state our results about the upper bound of low-dimensional structural generalization error with distribution shift. Notably, Eq. (11) holds for every real value $\lambda$, which is used to further analyze the generalization of BIQA models in Theorem 3.

**Theorem 3.** *Follow the notation in Lemma 4, and let $D\big(P_{test}\|P_{train}\big)$ denote the chi-square divergence between the training and test distributions. Let $L_\ell$ be the Lipschitz constant of the loss function $\ell$, and let $L$ be the number of layers. Then, for the BIQA model class $\mathcal{F}_L$, with probability at least $1-\delta$, for all $f \in \mathcal{F}_L$ simultaneously, we have:*

$$\mathrm{err}_P(f) \leq \mathrm{err}_S(f) + \mathcal{O}\left(\frac{L_\ell M \sqrt{L\left(D\left(P_{test}\|P_{train}\right)h(\Sigma)+1\right)}}{\sqrt{n}}\right) + c\sqrt{\frac{\log(1/\delta)}{2n}}. \quad (12)$$

*where $c$ is the upper bound of the $L_1$ loss, and $h(\Sigma)$ is the intrinsic dimension of low level features.*

*Proof.* Please refer to the Appendix H. $\qquad\square$

Beyond the conclusions derived from Theorems 1 and 2, Theorem 3 provides a much tighter bound on depth $L$, and Eq. (12) can further demonstrate that: **(1)** Theorem 3 provides a theoretical guarantee that the greater the distribution difference, the worse the generalization performance. **(2)** The generalization boundary is linearly and positively correlated to the Lipschitz constant of the BIQA loss function, suggesting that the choice of loss function has a significant influence on the generalization capability of the BIQA model. **(3)** The greater the value of $M = \prod_{i=1}^{L} M_F(i)$, the larger the upper bound, which implies that an excessively large number of parameters may diminish the generalization ability of the BIQA model. **(4)** The generalization bound exhibits a positive correlation with *Intrinsic dimension* of low-level quality feature, which theoretically reinforces its critical role in BIQA.

## 4 REPRESENTATION CAPACITY BOUND OF BIQA MODELS

To investigate what governs the upper bound of quality-aware representation capacity in BIQA, we examine the posterior distribution of the BIQA model to characterize its representation capacity.

Specifically, as established in (McAllester, 1999), models that simultaneously maintain low empirical risk and posterior distributions near the prior distribution tend to demonstrate reduced expected error. Thus, we use the Probably Approximately Correct Bayesian (PAC-Bayes) (Langford & Seeger, 2001) theorem to bound the representation capacity of BIQA models. To better illustrate our theoretical insights, we consider the form of Kullback–Leibler (KL) divergence for PAC-Bayes in Lemma 6.

**Lemma 6** (PAC-Bayes, KL divergence form (Dziugaite & Roy, 2017))**.** *Let $\delta \in (0, 1)$, $n \in \mathbb{N}$ and $\mu$ be a data-generating distribution over $\mathbb{R}^k$. Let $P$ be any prior distribution over a hypothesis space $\mathcal{H}$. Let $S_n = \{(x_i, y_i)\}_{i=1}^n$ be an i.i.d. sample drawn from $\mu$. Then, with probability at least $1-\delta$ over the choice of $S_n \sim \mu^n$, for every posterior distribution $Q$ on $\mathcal{H}$, the following inequality holds:*

$$\mathrm{KL}\left(\mathbb{E}_{f\sim Q}\left[\mathrm{err}_{S_n}(f)\right] \big\| \mathbb{E}_{f\sim Q}\left[\mathrm{err}_\mu(f)\right]\right) \leq \frac{\mathrm{KL}(Q\|P) + \ln\left(\frac{n}{\delta}\right)}{n-1}, \quad (13)$$

*where $\mathrm{err}_{S_n}(f)$ is the empirical risk of $f$ on the sample $S_n$, and $\mathrm{err}_\mu(f)$ is the expected risk of $f$ under the true distribution $\mu$. The expectations are taken over hypotheses $f \sim Q$. $\mathrm{KL}(\cdot\|\cdot)$ is the Kullback–Leibler divergence between Bernoulli distributions with respective mean values.*

Although it is widely believed that the representation capacity of a CNN-based blind image quality assessment (BIQA) model is primarily determined by its model dimension $d$, this assumption lacks solid theoretical justification. Current understanding relies mainly on the empirical belief that a larger number of trainable parameters enables the network to represent a richer class of functions. However, from the perspective of PAC-Bayes theory, as shown in Eq. (13) of Lemma 6, the quality representation capacity is fundamentally reflected by the KL divergence $\mathrm{KL}(Q\|P)$, which measures the discrepancy between the posterior and prior distributions. This is because a more expressive model tends to induce a posterior distribution that deviates further from the prior to better fit the training data, resulting in a higher KL value. Consequently, we can rigorously characterize the representation capacity of a CNN-based BIQA model by analyzing the upper bound of $\mathrm{KL}(Q\|P)$. Our theoretical findings also provide formal justification for the aforementioned belief—demonstrating that a larger model dimension $d$ indeed serves as a valid indicator of enhanced representation capacity in BIQA.

Now, we aim to **estimate the representation capacity via PAC-Bayes**. Based on Lemma 6, we use the Kullback–Leibler divergence term $\mathrm{KL}(Q\|P)$ to characterize the representation capacity of our CNN-based model. Intuitively, a larger divergence between the prior distribution $P$ and the posterior distribution $Q$ implies that the posterior can capture a broader class of functions, indicating

stronger expressive capacity. From this perspective, we are free to assume a relatively simple prior distribution, while calculating the posterior is significantly complex. This idea is aligned with recent studies (Dziugaite & Roy, 2017; Izmailov et al., 2021), which provide qualitative analyses of the structure and expressivity of posterior distributions in Bayesian neural networks (BNNs). Given the closure properties of Gaussian distributions under affine transformations, we adopt the assumption:

**Assumption 1** (Gaussian Prior and Posterior for BCNNs). *For Bayesian convolutional neural networks (BCNNs), we assume that the prior and posterior distributions both belong to the multivariate Gaussian: $P = \mathcal{N}(0, I_n)$, where $I_n$ is the identity covariance matrix; $Q = \mathcal{N}(\mu, \Sigma)$, where $\mu \in \mathbb{R}^n$ is a learned mean vector and $\Sigma \in \mathbb{R}^{n \times n}$ is a learned covariance matrix.*

Under this assumption, we can provide a PAC-Bayesian estimate for the representation capacity of the BIQA model in terms of the KL divergence.

**Theorem 4** (KL Divergence Bound for CNN-based BIQA models). *Let $P = \mathcal{N}(0, I)$ and $Q = \mathcal{N}(\mu, \Sigma)$ be the prior and approximate posterior distributions for a CNN-based BIQA model in $\mathbb{R}^d$, respectively. Suppose this CNN model has $L$ layers, where the $l$-th layer has $m_l$ filters and $d_l$ parameters, and the kernel size of each filter is $k$. Then the total number of parameters is $\sum_{l=1}^{L} d_l = k \sum_{l=1}^{L} m_l$. If the output MOS scores are constrained in $[0, r]$ for some $r > 0$, then:*

$$\mathrm{KL}(Q\|P) \leq \mathcal{O}\Big(k\Big(\sum_{l=1}^{L} m_l\Big)r^2\Big). \tag{14}$$

*Proof.* Please refer to the Appendix I. □

Through Theorem 4, we can make the following conclusions: **(1)** As the depth $L$ and kernel size $k$ increase, the $\mathrm{KL}(Q\|P)$ grows. This indicates that learning high-level quality perception features has a positive impact on the representation capacity of BIQA models, allowing them to better fit the training data and achieve smaller empirical errors. Our experiments show that this theoretical result in Eq. (14) is consistent with our empirical observations on different datasets. **(2)** In BIQA models, the MOS range $r$ exhibits positive correlations with both representation capacity and generalization capability (Theorem 2), though its correlation with representation capacity appears more pronounced. This relationship may stem from the enhanced learning precision enabled by wider MOS ranges.

## 5 DISCUSSIONS

In this section, we first describe the conflict between robust quality-aware generalization and representation capacity in BIQA models based on the theoretical analysis in Sections 3 and 4. Then, we show how the theoretical findings inspire a global and theoretical explanation for the good performance of existing BIQA methods. Based on these discussions, we finally provide suggestions for improving the BIQA performance in Appendix J.

### 5.1 CONFLICT OF STRONG REPRESENTATION AND GENERALIZATION IN BIQA MODELS

In addition to the conclusions in Sections 3 and 4, these theorems yield further insights. As discussed in these sections, when the level of quality perception feature increases, (1) the generalization error bound of BIQA models increase (see Theorems 1, 2 and 3); and (2) the representation capacity tends to increase because of the greater KL divergence bound (see Theorem 4). Consequently, for BIQA models with a limited number of hidden units, emphasizing only high-level quality perception features is not necessarily beneficial, as there is a clear trade-off between achieving a good generalization bound and having strong representation capacity. Specifically, we conclude that as the level of the quality perception feature increases, the test error of BIQA networks may first decrease and then increase. This outcome is proved in Section 6 of the experiments.

### 5.2 THEORETICAL EXPLANATION FOR EXISTING IQA MODELS

In current DL-based IQA models, various methods have been proposed to improve generalization. They can be broadly categorized as follows: training with **(1)** extra datasets or network branches (for example, CONTRIQUE (Madhusudana et al., 2022) and LIQA (Zhang et al., 2022a)); **(2)** superior loss functions (such as NIMA (Talebi & Milanfar, 2018) and Norm-in-Norm (Li et al., 2020)); and

**(3)** effective feature fusion (such as MUSIQ (Ke et al., 2021), Hyper-IQA (Su et al., 2020), and Stair-IQA (Sun et al., 2022)). The benefit of approaches in Category **(1)** is evident; it parallels the effect of increasing the data size $n$ in Eqs. (6-7,9,12-13). For Category **(2)**, since the Lipschitz constant $L_\ell$ of the loss function greatly affects the generalization bound (Eq. (12) in Theorem 3), a suitable loss function can help improve generalization. For Category **(3)**, according to the analysis in Section 5.1, low-level and high-level image features complement one another, so multi-level image quality feature learning and fusion can improve generalization. Moreover, in BIQA models using test-time adaptation (TTA) (Roy et al., 2023), generalization is improved by reducing distribution shifts between training and testing sets by unsupervised fine-tuning, which aligns with Theorem 3.

## 6 EXPERIMENTS

We empirically validate the theoretical results presented earlier through extensive experiments. More experimental results are shown in Appendix L.1 to prove the practical values of our theorems. Details about the datasets and experimental settings used in this study are provided in the Appendix L.2.

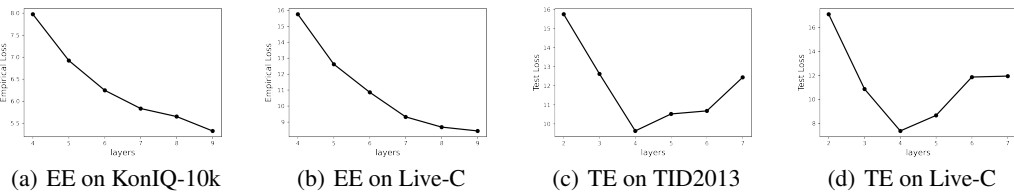

|   (a) EE on KonIQ-10k   |   (b) EE on Live-C   |   (c) TE on TID2013   |   (d) TE on Live-C   |

Figure 1: Impact of network depth on empirical error (EE) and test error (TE).

**Experimental Verification of Theorem 4**    As stated in Conclusion (1) of Theorem 4, the empirical error of the BIQA model decreases when the network depth and the level of quality features increase. To examine this, we train a CNN (based on the official demo) with different depths $L$ and a restricted number of hidden units on KonIQ-10k (Hosu et al., 2020) and Live-C (Ghadiyaram & Bovik, 2015), extracting quality perception features in the $(L-1)$-th layer. Figures 1(a-b) show that deeper BIQA networks, which learn higher-level quality perception features, produce smaller empirical errors than shallower networks with lower-level features. These observations confirm Theorem 4.

**Experimental Verification of Theorems 1, 2, 3 and Discussion Results of Section 5.1**    According to Theorems 1, 2 and 3, the generalization ability of a BIQA model decreases as $L$ grows. In Section 5.1, we propose the conjecture that test error may first decrease and then increase with depth. To test this, we use the same CNN setup as before with different depths $L$ and a restricted number of hidden units, training on KonIQ-10k (Hosu et al., 2020) and testing on TID2013 (Ponomarenko et al., 2015) and Live-C (Ghadiyaram & Bovik, 2015). Figures 1(c-d) verify that the test error follows the pattern predicted by Theorems 1, 2, 3 and Section 5.1.

## 7 CONCLUSION

From a theoretical standpoint, this work innovatively investigates the theoretical upper bounds of the generalization ability and representation capacity of DL-based BIQA models. First, we propose three different theorems to establish and prove generalization bounds for BIQA models under scenarios with or without distribution shifts between training and test data, highlighting the importance of low-level quality features. Then, we demonstrate the significance of high-level features on the representation capacity. These findings reveal a conflict between the generalization capability and the representation capacity of BIQA models. In response, we provide practical suggestions and theoretical supports for existing BIQA training, proving the practical values of our theorems. To our knowledge, this is the first work in IQA that offers the theoretical upper bounds for generalization error and representation capacity, addressing a key gap in the theoretical study in the IQA domain.

ETHICS STATEMENT

Our research on image quality assessment is centered around theoretical derivation, and all experiments were conducted on publicly available, anonymized benchmark datasets. This study did not involve human or animal subjects, and we foresee no direct negative societal impacts or ethical concerns arising from our work.

REPRODUCIBILITY STATEMENT

To ensure reproducibility, clear explanations and complete proof of the claims and theorems are included in the appendix. The experimental code was based on the official CNN demo, which will be made available in a public repository upon publication. The datasets used are standard public benchmarks.

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

## A The Use of Large Language Models (LLMs)

In this work, we utilized LLMs solely for the purpose of English language refinement. These models were employed to assist with the proofreading and enhancement of written text, ensuring clarity, coherence, and grammatical accuracy. The LLMs were not used for generating content, and all research, analysis, and conclusions presented are the result of our own work and independent thought.

## B Related Works

### B.1 Blind Image Quality Assessment (BIQA)

BIQA has gained significant attention recently due to the absence of reference images in realistically distorted image datasets (Zhai & Min, 2020). With the development and wide applications of Deep Learning, numerous DL-based approaches have achieved notable progress in BIQA (Ke et al., 2021; Golestaneh et al., 2022), such as RankIQA (Liu et al., 2017), CONTRIQUE (Madhusudana et al., 2022), and GraphIQA (Simeng et al., 2023). Although these works enhance quality perception ability on training datasets, their generalization is restricted by the limited size of existing IQA datasets. To address this limitation, some methods seek to improve the generalization of BIQA through more complex modules (Lin et al., 2020; Ma et al., 2017; Zhou et al., 2022) or unsupervised pre-training strategies (Prabhakaran & Swamy, 2023; Saha et al., 2023).In order to handle the domain shift between training and test sets, one recent approach integrates domain adaptation and ensemble learning into the IOA task (Roy et al., 2023). In recent advances in the field of IQA, some new studies are proposed to improve the generalization or robustness of IQA models by combining BIQA with various learning paradigms adapted to specific scenarios (Zhang et al., 2024; Wang et al., 2023; Yang et al., 2024; Zhang et al., 2022a; Wang & Ma, 2021; Wang et al., 2021; Zhang et al., 2022b) such as Contrastive Learning, Continual Learning, Active Learning, Curriculum Learning, Multi-task Learning and Adversarial Learning. Although existing IQA methods achieve strong performances in different settings and scenarios, the improvements in generalization usually come with increased training costs (Zhang et al., 2024; Zhong et al., 2024). while the importance of representation learning for multi-level image features and distortion information is widely acknowledged in promoting IQA generalization (Ke et al., 2021; Su et al., 2020; Sun et al., 2022), theoretical guarantees remain elusive. Currently, there are no intuitive theoretical results addressing the generalization ability of BIQA models in existing literature.

### B.2 Generalization Bound in Deep Learning

Commonly used classification objectives in deep learning (such as cross-entropy loss) encourage a larger output margin, which is the gap between the predicted true label and the next most confident label. These ideas appeared before deep learning and have strong statistical guarantees for linear and kernel methods (Bartlett & Mendelson, 2002; Koltchinskii & Panchenko, 2002; Hofmann et al., 2008; Kakade et al., 2008) which help explain the success of algorithms like SVM (Boser et al., 1992; Cortes, 1995). Nevertheless, deep learning reveals statistical patterns that challenge conventional understanding (Zhang et al., 2021; Neyshabur et al., 2017), providing new perspectives for studying its generalization. These perspectives include insights into implicit and algorithmic regularization mechanisms (Soudry et al., 2018; Li et al., 2018), recent investigations of interpolation-based classifiers (Hastie et al., 2022; Bartlett et al., 2020), and analyses of the noise dynamics and the stability of stochastic gradient descent (SGD) (Keskar et al., 2016; Chaudhari et al., 2019). More recently, the generalization boundary of Multilayer Perceptrons (MLPs) has been established in the algorithm selection tasks (Wu et al., 2024b;a). Most existing work on theoretical generalization focuses on fully connected neural networks. Although some approaches (Zhou & Feng, 2018; Long & Sedghi, 2019) have examined the generalization of CNNs in recent years, they generally target classification rather than regression tasks. A few works (Jakubovitz et al., 2019; Amjad et al., 2021) discuss the expected error bound for regression networks, but they do not address IQA-specific characteristics. Our theoretical analysis fully considers the distinct properties of IQA, thus narrowing the theoretical gap in this area. We further discuss the broader applicability of our analysis to other regression tasks in Appendix M.

## C    THE PROOF OF LEMMA 1

We first define the Lipschitz condition.

**Definition 7** ((Valentine, 1945)). *A loss function $l$ is Lipschitz (with respect to its first argument) if there exists a constant $L > 0$ such that, for any $x_1, x_2$ (belonging to the domain of the first argument of $l$) and any fixed value $y$ (for the other arguments of $l$, if any), the following inequality holds:*

$$|l(x_1, y) - l(x_2, y)| \leq L|x_1 - x_2| \tag{15}$$

*where $L$ is named as the Lipschitz constant.*

Now we provide the formal proof of Lemma 1.

*Proof.* To prove that the $L_1$ loss function satisfies the Lipschitz condition, let $\mathbf{x}, \mathbf{y}, \mathbf{z} \in \mathbb{R}^n$ be arbitrary vectors. The $L_1$ loss function between $\mathbf{x}$ and $\mathbf{y}$ is defined as:

$$L_1(\mathbf{x}, \mathbf{y}) = \sum_{i=1}^{n} |x_i - y_i|. \tag{16}$$

To show that $L_1$ satisfies the Lipschitz condition, we need to find a constant $K \geq 0$ such that

$$|L_1(\mathbf{x}, \mathbf{y}) - L_1(\mathbf{x}, \mathbf{z})| \leq K \cdot \|\mathbf{y} - \mathbf{z}\|_1, \tag{17}$$

where $\|\mathbf{y} - \mathbf{z}\|_1 = \sum_{i=1}^{n} |y_i - z_i|$ is the $L_1$ norm. Consider the absolute difference in $L_1$ loss:

$$|L_1(\mathbf{x}, \mathbf{y}) - L_1(\mathbf{x}, \mathbf{z})| = \left| \sum_{i=1}^{n} |x_i - y_i| - \sum_{i=1}^{n} |x_i - z_i| \right|. \tag{18}$$

By the triangle inequality for absolute values, we have

$$|a - b| \leq |a| + |b|, \tag{19}$$

which implies

$$||x_i - y_i| - |x_i - z_i|| \leq |(x_i - y_i) - (x_i - z_i)| = |y_i - z_i|. \tag{20}$$

Summing over all $i$, we obtain

$$|L_1(\mathbf{x}, \mathbf{y}) - L_1(\mathbf{x}, \mathbf{z})| \leq \sum_{i=1}^{n} |y_i - z_i| = \|\mathbf{y} - \mathbf{z}\|_1. \tag{21}$$

Therefore, the $L_1$ loss satisfies the Lipschitz condition with $K = 1$. $\qquad\square$

## D    THE PROOF OF THEOREM 1 AND COROLLARY 1

### D.1    PROOF OF THEOREM 1

Here we provide the proof of our generalization bound for CNN-based BIQA models in Theorem 1.

*Proof.* For the $l$-th layer, the parameter space is a subset of $\mathbb{R}^{k_l d_l}$ (each filter has $k_l$ parameters, and there are $d_l$ filters). Standard estimates (via a grid argument on the ball of radius $r$) yield that for any $\epsilon > 0$ there exists an $\epsilon$–cover of the parameter space of that layer with cardinality at most

$$\mathcal{N}_l(\epsilon) \leq \left( \frac{a \cdot C}{\epsilon} \right)^{k_l d_l}, \tag{22}$$

where $C > 0$ is a universal constant. This follows by covering a ball in $\mathbb{R}^{k_l d_l}$ with Euclidean balls of radius $\epsilon$; note that the metric on the parameters is typically the Euclidean one.

Suppose that the model in the CNN-based BIQA network class $\mathcal{F}_L$ is a composition of $L$ layers. If each layer map is Lipschitz (which one shows under mild assumptions on the activation functions and

pooling operations), then after re–scaling if necessary the covering number of the composite function class $\mathcal{F}$ can be bounded by concatenating the covers for the individual layers. Hence one obtains:

$$\mathcal{N}(\epsilon, \mathcal{F}_L) \leq \prod_{l=1}^{L} \mathcal{N}_l(\epsilon_l). \tag{23}$$

For an error budget allocation where the cumulative errors across all layers are bounded by $\epsilon$, a careful telescoping selection of layer-wise tolerances $\epsilon_l \propto \epsilon$ (or through optimal allocation) yields the desired bound:

$$\log \mathcal{N}(\epsilon, \mathcal{F}_L) \leq \sum_{l=1}^{L} k_l d_l \log \left( \frac{a}{\epsilon} \right), \tag{24}$$

Standard results in learning theory show that, for any $f \in \mathcal{F}_L$, if the true risk $\mathrm{err}_P(f)$ (e.g., in a regression loss in BIQA training) satisfies a uniform concentration inequality based on the covering number of $\mathcal{F}_L$, then with probability at least $1 - \delta$ one has:

$$\mathrm{err}_P(f) \leq \mathrm{err}_S(f) + \mathcal{O}\left( \sqrt{\frac{\log \mathcal{N}(\epsilon, \mathcal{F}_L, \|\cdot\|_\infty) + \log(1/\delta)}{n}} \right), \tag{25}$$

Often, the choice of $\epsilon$ is optimized or absorbed into the constant inside the logarithm, so one may simply write:

$$\mathrm{err}_P(f) \leq \mathrm{err}_S(f) + \mathcal{O}\left( \sqrt{\frac{\sum_{l=1}^{L} k_l d_l \log(a/\epsilon) + \log(1/\delta)}{n}} \right). \tag{26}$$

Q.E.D $\hfill\square$

### D.2 THE PROOF OF COROLLARY 1

Here, we aim to prove the tighter upper bound for the generalization error based on *Covering number* for CNN-based BIQA models. We first introduce the following Lemma:

**Lemma 7** (Covering number of deep neural networks (Shen, 2024))**.** *Consider the class of deep neural networks*

$$\mathcal{F} := \mathcal{F}(1, d_0, d_1, \ldots, d_L, a), \tag{27}$$

*which is parameterized by $\theta \in [-a, a]^S$. Let the vector $(d_0, d_1, ..., d_L)$ represent the dimensions of the layers of the neural network $f(x; \theta) \in \mathcal{F}$. Suppose the radius of the domain $\mathcal{X}$ of $f \in \mathcal{F}$ is bounded by $a_x$ for some $a_x > 0$, and the activations $\phi_1, \ldots, \phi_l$ are 1-Lipschitz. Then for any $\epsilon > 0$, the covering number $\mathcal{N}(\mathcal{F}, \epsilon, \|\cdot\|_\infty)$ is bounded by*

$$\frac{(4(L+1)(a_x+1)(2a)^{L+2}(\prod_{j=0}^{L} d_j) \cdot \epsilon^{-1})^S}{d_1! \cdot d_2! \cdots d_L!}, \tag{28}$$

*where $S = \sum_{i=0}^{L} d_i d_{i+1} + d_{i+1}$. Especially, the range of the $(i-1)$-th hidden layer is bounded by $[-a^{(i)}, a^{(i)}]$ with*

$$a^{(i)} \leq (2a)^i \prod_{j=1}^{i} d_j, \quad for\ i = 1, \ldots, L. \tag{29}$$

Lemma 7 can be combined with the standard Dudley entropy integral upper bound on Rademacher complexity (see e.g. (Mohri et al., 2012)). Then we give the formal proof of Corollary 1:

*Proof.* Recall that for any $f \in \mathcal{F}_L$, with probability at least $1 - \delta$, one has

$$\mathrm{err}_P(f) \leq \mathrm{err}_S(f) + \mathcal{O}\left( \sqrt{\frac{\log \mathcal{N}(\epsilon, \mathcal{F}_L, \|\cdot\|_\infty) + \log(1/\delta)}{n}} \right). \tag{30}$$

From the Lemmaa 7, the *Covering number* for the network class is bounded by

$$\log \mathcal{N}(\epsilon, \mathcal{F}_L, \| \cdot \|_\infty) \le S \log \left( \frac{4(L+1)(a_x+1)(2a)^{L+2} \left( \prod_{j=0}^L d_j \right)}{\epsilon} \right) - \sum_{l=1}^L \log(d_l!), \quad (31)$$

where $S = \sum_{i=0}^L \left( d_i d_{i+1} + d_{i+1} \right)$.

Substitute the above logarithmic bound into the generalization error inequality. Then, with probability at least $1 - \delta$, every $f \in \mathcal{F}_L$ satisfies:

$$\mathrm{err}_P(f) \le \mathrm{err}_S(f) + O\left( \sqrt{ \frac{S \log \left( \frac{4(L+1)(r_x+1)(2r)^{L+2} \left( \prod_{j=0}^L d_j \right)}{\epsilon} \right) - \sum_{l=1}^L \log(d_l!) + \log(1/\delta)}{n} } \right) \tag{32}$$

Roughly, the generalization error is approximately bounded by

$$\mathrm{err}_P(f) \le \mathrm{err}_S(f) + \mathcal{O}\left( \sqrt{ \frac{L^2 \log(a/\epsilon) + \log(1/\delta)}{n} } \right). \tag{33}$$

Q.E.D $\hfill \square$

This estimation indicates that, roughly, the generalization gap grows with the square of the number of layers $L$ (up to a logarithmic factor in $a$) and decreases inversely with the square root of the sample size n, while also depending logarithmically on the failure probability $\delta$.

## E   THE PROOF OF LEMMA 2

*Proof.* From Sauer's Lemma (Sauer, 1972), which provides a fundamental bound on the growth function of a hypothesis class $\mathcal{H}$ for CNNs - specifically limiting the number of possible dichotomies (i.e., splits or labelings) realizable on any $n$ points. The lemma states that when the VC dimension $d_{vc}$ of $\mathcal{H}$ satisfies $n \ge d_{vc}$, then:

$$\Pi_{\mathcal{H}}(n) \le \sum_{i=0}^{d_{vc}} \binom{n}{i}. \tag{34}$$

Here, $\Pi_{\mathcal{H}}(n)$ is the growth function—the maximum number of ways $\mathcal{H}$ can label any set of n points. To simplify this sum, we use a standard bound on the binomial coefficients. In particular, for each $i$, we have $\binom{n}{i} \le \left( \frac{en}{i} \right)^i$, where $e$ is the base of the natural logarithm (approximately 2.718). Since the worst-case (largest) term in the sum occurs when $i = d$, the entire sum can be bounded by:

$$\Pi_{\mathcal{H}}(n) \le \sum_{i=0}^{d_{vc}} \left( \frac{en}{i} \right)^i \le \left( \frac{en}{d_{vc}} \right)^{d_{vc}}, \tag{35}$$

Through the application of a concentration inequality (in the spirit of Hoeffding's inequality) coupled with a union bound over all possible $\Pi_{\mathcal{H}}(2n)$ labelings realizable on the combined sample, one obtains the standard generalization bound (as formalized in Theorem 3.3 of (Anthony & Bartlett, 1999)):

$$\mathbb{P}\left( \sup_{h \in \mathcal{H}} \left| \mathrm{err}_P(f) - \mathrm{err}_S(f) \right| \ge \epsilon \right) \le 4\Pi_{\mathcal{H}}(2n) \exp\left( -\frac{n\epsilon^2}{8} \right). \tag{36}$$

Set the right–hand side equal to $\delta$:

$$4 \left( \frac{2en}{d_{vc}} \right)^{d_{vc}} \exp\left( -\frac{n\epsilon^2}{8} \right) = \delta. \tag{37}$$

Solving for $\epsilon$ yields

$$\epsilon \leq \sqrt{\frac{8}{n}\left(d_{\text{vc}}\log\frac{2en}{d_{\text{vc}}} + \log\frac{4}{\delta}\right)}. \tag{38}$$

Therefore, for any $\delta \in (0,1)$, with probability at least $1-\delta$ over an i.i.d. sample of $n$ samples, every hypothesis $h \in \mathcal{H}$ (in our case, this is each CNN-based BIQA model) satisfies:

$$\text{err}_P(f) \leq \text{err}_S(f) + \sqrt{\frac{8}{n}\left(d_{\text{vc}}\log\frac{2en}{d_{\text{vc}}} + \log\frac{4}{\delta}\right)}. \tag{39}$$

where $\text{err}_P(f)$ is the true (population) risk, $\text{err}_S(f)$ is the empirical (training) risk, and $d_{\text{vc}}$ is the VC–dimension of the hypothesis class $\mathcal{H}$ (in our case, it is the class of CNN-based BIQA models). $\square$

## F  THE PROOF OF THEOREM 2

We adopt the lemma for the results of growth functions and upper bound on the VC dimension.

**Lemma 8** ((Sepliarskaia et al., 2024b)). *Consider a convolutional neural network (CNN) class* $\mathcal{H}(k, m_0, \ldots, m_L, r)$*, where $L$ is the number of layers, $m_i$ is the width of neural network in layer $i$, $k$ is the kernel size (number of parameters associated with the local receptive field in each unit), and $r$ is a bound on output range or an additional scaling parameter.*

*For the $l$-th layer, let*

$$W_l = \sum_{j=1}^{l} m_j\left(km_{j-1} + 1\right), \tag{40}$$

*denote the total number of parameters from the first layer up to layer $l$. Then for any integer $m > 0$,*

$$\Pi_{\mathcal{H}}(m) \leq 2^L \prod_{l=1}^{L}\left(\frac{2emrm_l l}{W_l}\right)^{W_l} \cdot 2\left(\frac{2emL}{W_L+1}\right)^{W_L+1}. \tag{41}$$

*and the VC dimension of $\mathcal{H}(k, m_0, \ldots, m_L, r)$ is bounded above by*

$$UB(d_{\text{VC}}) = L + 1 + 4\left(\sum_{l=1}^{L} W_l\right)\log_2\left(8er\sum_{l=1}^{L} m_l\right). \tag{42}$$

The proof of Lemma 8 is provided in (Sepliarskaia et al., 2024b), with only an adjustment in the parameter formulation. This result extends previous VC dimension bounds ((Sepliarskaia et al., 2024b; Kohler & Walter, 2023)) of CNN by including the kernel size $k$, network width $m_i$ in the parameter count $W_l$, reflecting that each unit in the $i$-th layer has $km_{i-1}$ weights plus one bias term. Based on Lemma 8, we provide the proof of our VC dimension theorem.

*Proof.* We begin with the given bound and Eq. (42) into the result in Lemma 2:

$$\text{err}_P(f) \leq \text{err}_S(f) + \sqrt{\frac{8}{n}\left(d_{\text{vc}}\log\frac{2en}{d_{\text{vc}}} + \log\frac{4}{\delta}\right)}, \tag{43}$$

So we get:

$$\text{err}_P(f) \leq \text{err}_S(f) + \mathcal{O}\left(\sqrt{\frac{kL^2\log(rL)\log\left(\frac{n}{kL^2\log(rL)}\right) + \log(1/\delta)}{n}}\right). \tag{44}$$

This is the big-$\mathcal{O}$ result in terms of $L$, $k$, $r$, $n$, and $\delta$.

On the other hand, we take Eq. (41) into Eq. (36), and we get:

$$\epsilon \leq \sqrt{\frac{8}{n} \log \left( \frac{4 \cdot 2^L \prod_{l=1}^{L} \left( \frac{4enrm_l l}{W_l} \right)^{W_l} \cdot 2 \left( \frac{4enL}{W_L+1} \right)^{W_L+1}}{\delta} \right)}. \tag{45}$$

Since we are told that $W_l = \sum_{j=1}^{l} m_j \left( k m_{j-1} + 1 \right)$, so after taking the product over $l = 1, \ldots, L$, we get:

$$\prod_{l=1}^{L} \left( \mathcal{O} \left( \frac{nr}{k} \right) \right)^{\mathcal{O}(kl)} = \exp \left( \mathcal{O} \left( k \sum_{l=1}^{L} l \right) \log \left( \frac{nr}{k} \right) \right) = \exp \left( \mathcal{O} \left( kL^2 \log \left( \frac{nr}{k} \right) \right) \right). \tag{46}$$

Thus, the bound becomes

$$\epsilon \leq \mathcal{O} \left( \sqrt{\frac{8}{n} \left( kL^2 \log \left( \frac{nr}{k} \right) + \log(1/\delta) \right)} \right). \tag{47}$$

This is also the desired Big-$\mathcal{O}$ upper bound in terms of $L$, $k$, $r$, $n$, and $\delta$. Thus we have the following:

$$\mathrm{err}_P(f) \leq \mathrm{err}_S(f) + \mathcal{O} \left( \min \left\{ \sqrt{\frac{kL^2 \log(rL) \log \left( \frac{n}{kL^2 \log(rL)} \right) + \log(1/\delta)}{n}}, \right. \right.$$
$$\left. \left. \sqrt{\frac{8 \left( kL^2 \log \left( \frac{nr}{k} \right) + \log(1/\delta) \right)}{n}} \right\} \right). \tag{48}$$

Q.E.D $\hfill\square$

## G  THE PROOF OF LEMMA 5

*Proof.* We write the low level quality feature as: $\mathcal{F}_1(x) \in \mathbb{R}^d$ with covariance $\Sigma$:

$$\Sigma = \mathbb{E} \left[ (\mathcal{F}_1(x) - \mu)(\mathcal{F}_1(x) - \mu)^\top \right] \quad \text{with} \quad \mu = \mathbb{E} \left[ \mathcal{F}_1(x) \right]. \tag{49}$$

From the Definition 6, we first assume that the random feature $\mathcal{F}_1(x)$ actually lives in an $h$-dimensional linear subspace of $\mathbb{R}^d$. In other words, there exists an $h$ with $1 \leq h \leq d$ such that the support of $\mathcal{F}_1(x)$ is contained in an $h$-dimensional subspace. In that case, the covariance matrix $\Sigma$ has rank $h$ and we can write its spectral (eigenvalue) decomposition as

$$\Sigma = U \operatorname{diag}(\lambda_1, \lambda_2, \ldots, \lambda_h, 0, \ldots, 0) U^\top, \tag{50}$$

with $\lambda_1 \geq \lambda_2 \geq \cdots \geq \lambda_h > 0$. By definition, the spectral norm is $\|\Sigma\|_2 = \lambda_1$, and the trace is $\operatorname{tr}(\Sigma) = \sum_{i=1}^{h} \lambda_i$.

Now, consider the particularly nice case in which the variance is equally distributed among the $h$ active directions. That is, assume $\lambda_1 = \lambda_2 = \cdots = \lambda_h$. Then one immediately obtains $\operatorname{tr}(\Sigma) = h\lambda_1$, and hence $h(\Sigma) = \operatorname{tr}(\Sigma)/\|\Sigma\|_2 = h\lambda_1/\lambda_1 = h$.

Thus, when the feature is exactly $h$-dimensional (and the data is isotropically spread among those directions), the effective dimension measured by $h\Sigma$) coincides with the actual dimension $h$. $\hfill\square$

In practice, especially for CNN features, the output $\mathcal{F}_1(x)$ may lie in $\mathbb{R}^d$ with $d$ large, but the significant variability is often concentrated in only a few directions.

## H   THE PROOF OF THEOREM 3

We first give the definition of Rademacher Average.

**Definition 8** ((Kakade et al., 2008)). *Suppose $\mathcal{F} : \mathcal{X} \to \mathbb{R}$ is a model space with a single dimensional output. The Rademacher Average (RA) (also known as Rademacher Complexity) of $\mathcal{F}$ is defined as follows:*

$$\mathcal{R}_n(\mathcal{F}) = \mathbb{E}_{\mathbf{x},\sigma} \left[ \sup_{f \in \mathcal{F}} \frac{1}{n} \sum_{i=1}^{n} f(x_i) \sigma_i \right] \tag{51}$$

*where $\sigma_i$ independently takes values in $\{+1, -1\}$ with equal probability. $\mathbf{x} = \{x_1, \cdots, x_n\} \sim P_x^n$.*

Now we provide the formal proof of Theorem 3.

*Proof.* According to the Definition 8 and considering the distribution shift in training and test sets in Lemma 4, the *Rademacher complexity* for $\mathcal{F}_L$ defined in Eqs. (1-3) can be upper bounded as:

$$R_{n,\boldsymbol{\eta}}(\mathcal{F}_L) \le \frac{1}{n\lambda} \log \left( 2^L \cdot \mathbb{E}_{\boldsymbol{\epsilon}} \exp \left( M\lambda \left\| \sum_{i=1}^{n} \epsilon_i \eta_i x_i \right\| \right) \right) \tag{52}$$

where $n$ denotes the number of training instances, $\mathbf{x}_i$ denotes the $i$-th instance, $\epsilon_i$ is a Rademacher variable, $\lambda$ is a random variable, $\eta_i = P_{\text{test}}(x_i)/P_{\text{train}}(x_i)$ is the defined importance weight for sample $x_i$, and $M$ satisfies that:

$$M = \prod_{j=1}^{L} M_F(j) \tag{53}$$

Let $Z = M \left\| \sum_{i=1}^{n} \epsilon_i \eta_i x_i \right\|$, as a random function of the $n$ Rademacher variables. Then we have:

$$R_{n,\boldsymbol{\eta}}(\mathcal{F}_L) \le \frac{1}{n} \frac{1}{\lambda} \log \left\{ 2^L \mathbb{E} \exp(\lambda Z) \right\} \tag{54}$$

Note that:

$$\frac{1}{\lambda} \log \left\{ 2^L \mathbb{E} \exp(\lambda Z) \right\} = \frac{L \log(2)}{\lambda} + \frac{1}{\lambda} \log\{ \mathbb{E} \exp \lambda(Z - \mathbb{E}[Z])\} + \mathbb{E}[Z] \tag{55}$$

For the third term in the right part of Eq. (55), by Jensen's inequality and Lemma 5, we have

$$\sum_{i=1}^{n} \eta_i^2 \|x_i\|^2 \le h(\Sigma) \|\Sigma\|_2 \sum_{i=1}^{n} \eta_i^2. \tag{56}$$

Substituting into the estimate for $\mathbb{E}[Z]$ we obtain

$$\mathbb{E}[Z] \le M \sqrt{h(\Sigma) \|\Sigma\|_2 \sum_{i=1}^{n} \eta_i^2}. \tag{57}$$

According to the property of Rademacher variable, we have:

$$Z(\epsilon_1, \ldots, \epsilon_i, \ldots, \epsilon_n) - Z(\epsilon_1, \ldots, -\epsilon_i, \ldots, \epsilon_n) \le 2M\eta_i \|x_i\|. \tag{58}$$

Hence, by standard results in bounded-difference condition (Boucheron et al.; 2013), the function $Z$ is sub-Gaussian with variance factor

$$\sigma^2 = \frac{1}{4} \sum_{i=1}^{n} \left( 2M\eta_i \|x_i\| \right)^2 = M^2 \sum_{i=1}^{n} \eta_i^2 \|x_i\|^2. \tag{59}$$

This implies that for any $\lambda > 0$

$$\frac{1}{\lambda} \log \left\{ \mathbb{E} \exp \left( \lambda(Z - \mathbb{E}[Z]) \right) \right\} \le \frac{\lambda M^2 \sum_{i=1}^{n} \eta_i^2 \|x_i\|^2}{2}. \tag{60}$$

Thus, we obtain

$$\frac{1}{\lambda} \log\{2^L \mathbb{E} \exp(\lambda Z)\} \leq \frac{L \log 2}{\lambda} + \frac{\lambda M^2 \sum_{i=1}^n \eta_i^2 \|x_i\|^2}{2} + \mathbb{E}[Z]. \tag{61}$$

Then, according to Eq. (56), we can write:

$$\frac{\lambda M^2 \sum_{i=1}^n \eta_i^2 \|x_i\|^2}{2} \leq \frac{\lambda M^2 h(\Sigma) \|\Sigma\|_2 \sum_{i=1}^n \eta_i^2}{2}. \tag{62}$$

The next step is to choose $\lambda > 0$ optimally to balance the two $\lambda$-dependent terms. That is, we want to minimize

$$f(\lambda) = \frac{L \log 2}{\lambda} + \frac{\lambda M^2 h(\Sigma) \|\Sigma\|_2 \sum_{i=1}^n \eta_i^2}{2}. \tag{63}$$

Taking the derivative with respect to $\lambda$ and setting it to zero gives

$$-\frac{L \log 2}{\lambda^2} + \frac{M^2 h(\Sigma) \|\Sigma\|_2 \sum_{i=1}^n \eta_i^2}{2} = 0, \tag{64}$$

or, equivalently,

$$\lambda = \sqrt{\frac{2L \log 2}{M^2 h(\Sigma) \|\Sigma\|_2 \sum_{i=1}^n \eta_i^2}}. \tag{65}$$

Plugging this optimal $\lambda$ back into Eq. (61), the sum of the first two terms becomes

$$\frac{L \log 2}{\lambda} + \frac{\lambda M^2 h(\Sigma) \|\Sigma\|_2 \sum_{i=1}^n \eta_i^2}{2} = M \sqrt{2L \log 2\, h(\Sigma) \|\Sigma\|_2 \sum_{i=1}^n \eta_i^2}. \tag{66}$$

Hence, the overall bound becomes

$$\frac{1}{\lambda} \log\{2^L \mathbb{E} \exp(\lambda Z)\} \leq M \sqrt{2L \log 2\, h(\Sigma) \|\Sigma\|_2 \sum_{i=1}^n \eta_i^2} + M \sqrt{h(\Sigma) \|\Sigma\|_2 \sum_{i=1}^n \eta_i^2}. \tag{67}$$

Thus, we finally obtain the bound

$$R_{n,\boldsymbol{\eta}}(\mathcal{F}_L) \leq \frac{M}{n} \sqrt{h(\Sigma) \|\Sigma\|_2 \sum_{i=1}^n \eta_i^2} \left(\sqrt{2L \log 2} + 1\right), \tag{68}$$

where $h(\Sigma) = \frac{\operatorname{tr}(\Sigma)}{\|\Sigma\|_2}$, $\quad 1 \leq h(\Sigma) \leq d$.

Following (Jeffreys, 1946), the chi-square divergence from the training distribution to the test distribution is given by

$$D\left(P_{\text{test}} \| P_{\text{train}}\right) = \int \frac{P_{\text{test}}^2(x)}{P_{\text{train}}(x)} dx - 1. \tag{69}$$

Note that $\eta_i = \frac{P_{\text{test}}(x_i)}{P_{\text{train}}(x_i)}$ is the the importance weight for data samples $\{x_i\}_{i=1}^n$ (drawn from $P_{\text{train}}$). Then the quantity $\frac{1}{n} \sum_{i=1}^n \eta_i^2 \|x_i\|^2$ as an empirical average approximates $\mathbb{E}_{x \sim P_{\text{train}}} \left[\frac{P_{\text{test}}^2(x)}{P_{\text{train}}^2(x)} \|x\|^2\right]$.

By the Law of Large Numbers (see, e.g., (Durrett, 2019)), in the limit $n \to \infty$ we have:

$$\begin{aligned}
\frac{1}{n} \sum_{i=1}^n \eta_i^2 \|x_i\|^2 &= \frac{1}{n} \sum_{i=1}^n \frac{P_{\text{test}}^2(x_i)}{P_{\text{train}}^2(x_i)} \|x_i\|^2 \\
&\approx \lim_{n \to +\infty} \frac{1}{n} \sum_{i=1}^n \frac{P_{\text{test}}^2(x_i)}{P_{\text{train}}^2(x_i)} \|x_i\|^2 = \mathbb{E}_{x_j \sim P_{\text{train}}} \left[\frac{P_{\text{test}}^2(x_j)}{P_{\text{train}}^2(x_j)} \|x_j\|^2\right].
\end{aligned} \tag{70}$$

Recall that the feature vector $x$ may in fact have high ambient dimension, but its significant variability is captured by its covariance matrix.

$$\Sigma = \mathbb{E}_{x \sim P_{\text{train}}} \left[ (x - \mu)(x - \mu)^\top \right], \qquad \mu = \mathbb{E}[x], \tag{71}$$

which leads to recognizing that the average squared norm $\frac{1}{n} \sum_{i=1}^n \|x_i\|^2$ is essentially $\text{tr}(\Sigma)$ and that the "per-direction" maximal magnitude is given by $\|\Sigma\|_2$, yielding the equivalent bound as:

$$\frac{1}{n} \sum_{i=1}^n \eta_i^2 \|x_i\|^2 \leq (D\left(P_{\text{test}} \| P_{\text{train}}\right) + 1) \frac{\text{tr}(\Sigma)}{\|\Sigma\|_2} + o\left(\frac{1}{\sqrt{n}}\right). \tag{72}$$

That is,

$$\frac{1}{n} \sum_{i=1}^n \eta_i^2 \|x_i\|^2 \leq (D\left(P_{\text{test}} \| P_{\text{train}}\right) + 1) h(\Sigma) + o\left(\frac{1}{\sqrt{n}}\right), \tag{73}$$

which appears as a multiplicative factor in the complexity term of the risk bound for the BIQA model. Then, substituting the bound (73) into the derivation of $R_{n,\boldsymbol{\eta}}(\mathcal{F}_L)$, we can obtain:

$$R_{n,\boldsymbol{\eta}}(\mathcal{F}_L) \leq \mathcal{O}\left( \sqrt{\frac{(D\left(P_{\text{test}} \| P_{\text{train}}\right) + 1) h(\Sigma) \cdot \Phi(n)}{n}} \right) + o\left(\frac{1}{\sqrt{n}}\right), \tag{74}$$

where $\Phi(n)$ denotes additional terms arising from, for example, the complexity of the function class $\mathcal{F}_L$ and other constants. In other words, the effective dimension $h(\Sigma)$ acts as an "amplifier" of the divergence between the test and training distributions in the risk bound.

Specifically, we can obtain:

$$\begin{aligned}
R_{n,\boldsymbol{\eta}}\left(\mathcal{F}_L\right) &\leq \frac{1}{n} \frac{1}{\lambda} \log \left\{ 2^L \mathbb{E} \exp(\lambda Z) \right\} \\
&\leq \frac{1}{\sqrt{n}} M(\sqrt{2\log(2)L} + 1) \sqrt{\sum_{i=1}^n \eta_i^2 \|x_i\|^2} \\
&= M(\sqrt{2\log(2)L} + 1) \sqrt{\frac{1}{n} \sum_{i=1}^n \eta_i^2 \|x_i\|^2} \\
&\leq M(\sqrt{2\log(2)L} + 1) \sqrt{\left( D\left(P_{\text{test}} \| P_{\text{train}}\right) + 1 \right) h(\Sigma) + o\left(\frac{1}{\sqrt{n}}\right)}.
\end{aligned} \tag{75}$$

Ignoring the lower-order term, it gives:

$$R_{n,\boldsymbol{\eta}}(\mathcal{F}_L) \leq \mathcal{O}\left( M(\sqrt{2\log(2)L} + 1) \sqrt{h(\Sigma)\left(D\left(P_{\text{test}} \| P_{\text{train}}\right) + 1\right)} \right). \tag{76}$$

Finally, we can substitute $R_{n,\boldsymbol{\eta}}\left(\mathcal{F}_L\right)$ into Lemma 3 and obtain the result in Theorem 3. Namely, for any BIQA model $f$ in the considered function class, we have

$$\text{err}_P(f) \leq \text{err}_S(f) + \mathcal{O}\left( \frac{L_l M(\sqrt{2\log(2)L} + 1)\sqrt{h(\Sigma)\left(D\left(P_{\text{test}} \| P_{\text{train}}\right) + 1\right)}}{\sqrt{n}} \right) + c\sqrt{\frac{\log(1/\delta)}{2n}}. \tag{77}$$

Since $\sqrt{2\log(2)L} + 1 = \mathcal{O}(\sqrt{L})$ for large $L$, we can simplify this as

$$\text{err}_P(f) \leq \text{err}_S(f) + \mathcal{O}\left( \frac{L_l M\sqrt{L\left(D\left(P_{\text{test}} \| P_{\text{train}}\right) h(\Sigma) + 1\right)}}{\sqrt{n}} \right) + c\sqrt{\frac{\log(1/\delta)}{2n}}. \tag{78}$$

Q.E.D $\hfill\square$

# I  THE PROOF OF THEOREM 4

*Proof.* From the Assumption 1, we are given the prior distribution: $P = \mathcal{N}(0, I)$, and the approximate (posterior) distribution produced by the CNN Bayesian network: $Q = \mathcal{N}(\mu, \Sigma)$ for CNN-based BIQA models. The Kullback–Leibler (KL) divergence between two multivariate Gaussian distributions $Q = \mathcal{N}(\mu, \Sigma)$ and $P = \mathcal{N}(0, I)$ in $d$ dimensions is given by:

$$
\mathrm{KL}(\mathcal{N}(\mu_0, \Sigma_0)\|\mathcal{N}(\mu_1, \Sigma_1)) = \frac{1}{2}\left(\mathrm{tr}(\Sigma_1^{-1}\Sigma_0) + (\mu_1 - \mu_0)^T\Sigma_1^{-1}(\mu_1 - \mu_0) - d + \ln\frac{\det\Sigma_1}{\det\Sigma_0}\right)
\tag{79}
$$

In our problem: $\mu_0 = \mu$ and $\Sigma_0 = \Sigma$ (from $Q$), $\mu_1 = 0$ and $\Sigma_1 = I$ (from $P$), thus we have:

$$
\mathrm{KL}(Q\|P) = \frac{1}{2}\left(\mathrm{tr}(\Sigma) + \mu^T\mu - d - \ln\det\Sigma\right)
\tag{80}
$$

Assume the CNN-based BIQA model has $L$ layers, the $l$-th layer has $m_l$ filters, and each filter's kernel size is $k$. Then, if each layer contributes $d_l = km_l(l = 1, \cdots, L)$ parameters, the total parameter count is $d = \sum_{l=1}^{L} d_l = k\sum_{l=1}^{L} m_l$. If the posterior covariance is diagonal with entries $\{\sigma_i^2\}_{i=1}^{d}$ and corresponding means $\{\mu_i\}_{i=1}^{d}$, we can write

$$
\mathrm{tr}(\Sigma^{-1}) = \sum_{i=1}^{d}\frac{1}{\sigma_i^2}, \quad \mu^T\Sigma^{-1}\mu = \sum_{i=1}^{d}\frac{\mu_i^2}{\sigma_i^2}, \quad \ln|\Sigma| = \sum_{i=1}^{d}\ln\sigma_i^2.
\tag{81}
$$

Hence, we have a Bayesian CNN where the parameters follow multivariate Gaussian distributions, in accordance with standard Bayesian deep learning methodology. Thus,

$$
\mathrm{KL}(Q\|P) = \frac{1}{2}\left[\sum_{i=1}^{d}\left(\sigma_i^2 + \mu_i^2\right) - d - \sum_{i=1}^{d}\ln\sigma_i^2\right].
\tag{82}
$$

Substituting the total number of parameters $d = k\sum_{l=1}^{L} m_l$ into Eq. (82), we have the final expression:

$$
\mathrm{KL}(Q\|P) = \frac{1}{2}\left[\sum_{i=1}^{k\sum_{l=1}^{L} m_l}\left(\sigma_i^2 + \mu_i^2\right) - k\sum_{l=1}^{L} m_l - \sum_{i=1}^{k\sum_{l=1}^{L} m_l}\ln\sigma_i^2\right].
\tag{83}
$$

Then, in the application of BIQA domain, we can assume that the output MOS scores of CNN-based BIQA models are forced to lie in a bounded range, say $[0, r]$, which implies that each mean satisfies:

$$
|\mu_{(l,i)}| \leq r \implies \mu_{(l,i)}^2 \leq r^2.
\tag{84}
$$

Then we obtain:

$$
\mathrm{KL}(Q\|P) \leq \frac{1}{2}\left[\sum_{i=1}^{d}\sigma_i^2 + dr^2 - d - \sum_{i=1}^{d}\ln\sigma_i^2\right],
\tag{85}
$$

We further assume that the contributions from $\sigma_i^2$ and $\ln\sigma_i^2$ are both bounded by positive constants (i.e., each term contributes $\mathcal{O}(1)$ per parameter). Then, $\sum_{i=1}^{d}\sigma_i^2 = \mathcal{O}(d)$, and $\sum_{i=1}^{d}\ln\sigma_i^2 = \mathcal{O}(d)$.

Thus, the term $dr^2$ dominates if $r$ is not just a fixed constant. Combining these above observations, we can obtain the representation capacity of CNN-based BIQA models with the format of KL divergence:

$$
\mathrm{KL}(Q\|P) \leq \mathcal{O}\left(k\left(\sum_{l=1}^{L} m_l\right)r^2\right).
\tag{86}
$$

Q.E.D $\qquad\qquad\qquad\qquad\qquad\qquad\qquad\qquad\qquad\qquad\qquad\qquad\qquad\qquad\qquad\qquad\square$

## J   EXEMPLARY SUGGESTIONS RELATED TO PROPOSED THEOREMS

The proposed theorems not only clarify existing work but also suggest further possibilities. Below are some practical recommendations for BIQA network design: **(1)** In the feature perspective, according to Conclusion (1) of Theorem 1 and Conclusion (1) of Theorem 4, low-level and high-level features are complementary. We thus suggest fusing multi-level features to handle distortion complexity and improve generalization. Figure 3 in Appendix L.3 shows such a strategy by integrating two levels of features. **(2)** In the loss perspective, based on Eq. (12) in Theorem 3, we propose to improve the loss function. For example, by incorporating a regularization term to minimize empirical error and enhance representation capacity without increasing network depth, the conflict between high-level and low-level image features can be avoided. **(3)** In terms of network parameters, according to Conclusion (3) of Theorem 3, limiting the size of parameters via another regularization term can be helpful. Detailed explanations for these suggestions are provided in Appendix K. Based on the Suggestions **(2)** and **(3)**, we propose the following exemplary loss function:

$$L_{BIQA} = L_1\left(f(x), y\right) + \mu \mathrm{Norm}\left\{\cos\left[R_B\left(x_f\right), R_B\left(y\right)\right]\right\} + \nu \|\mathbf{W}_L\|_F \tag{87}$$

where $x_f$ denotes the extracted quality perception feature vector, and $R_B\left(x_f\right)$ is the ordering (from smallest to largest) of Euclidean distances between $x_f$ and the feature vectors of other images in the same batch $B$. The functions $\cos$ and $\mathrm{Norm}$ denote the cosine similarity and the max-min normalization (with maximum value 1 and minimum value $\cos([1, 2, \ldots, B-1], [B-1, B-2, \ldots, 1])$), respectively. The second term (consistency regularizer) in Eq. (87) corresponds to Suggestion **(2)**, which can directly minimize empirical error by remaining the consistency between feature space and label space. The third term (parameter regularizer) corresponds to Suggestion **(3)**, which constrains the size of the final prediction layer's weight parameter $\mathbf{W}_L$ using the Frobenius norm $\|\cdot\|_F$. $\mu$ and $\nu$ are the hyper-parameters. More details on Eq. (87) are given in Appendix L.4.

## K   THEORETICAL EXPLANATION FOR THREE SUGGESTIONS

For Suggestion **(1)**, according to Theorems 1, 2 and 3, as the level of image feature decreases, the generalization bound also decreases, and the generalization for quality perception becomes more excellent, which illustrates the significant role of low-level image features. According to Theorem 4, as the level of image feature increases, the greater representation capacity of BIQA network tends to increase due to the growing $\mathrm{KL}(Q\|P)$, which illustrates the important role of high-level image features. Therefore, Suggestion **(1)** is theoretically valid.

For Suggestion **(2)**, on the one hand, since the Lipschitz constant $L_\ell$ of the loss function is strongly related to the generalization bound, as shown in Eq. (12) in the proof of Theorem 3, an appropriate loss function can facilitate better generalization. Similar to the methods described in Category **(2)** in Section 5.2, Suggestion **(2)** promotes better generalization by improving the loss function with one regular term in Eq. (87). On the other hand, according to the analysis in Section 3 and Section 4, we can conclude that there exists a conflict between good generalization and strong representation for BIQA networks with a restricted number of hidden units. In other words, when attempting to enhance the representation capacity of a BIQA network by increasing the learning level of quality perception features for MOS prediction, the model might have to incur the cost of weaker generalization capacity. This naturally leads to the following question: *Can we reduce expected error and enhance the representation capacity of a BIQA network without increasing the complexity of learning quality perception features through an alternative approach?* The answer is yes, and the regularization term in Suggestion **(2)** is a typical approach, which can enhance the representation capacity (i.e., reduce expected error) and keep good generalization simultaneously. Therefore, Suggestion **(2)** is theoretically valid.

For Suggestion **(3)**, according to the theoretical result in Eqs. (11-12) in Theorem 3, we can observe that a tighter weight parameter boundary $M$ may promote better generalization. This motivates us to apply a regularization penalty on the model parameters during the training process. Therefore, Suggestion **(3)** is theoretically valid.

Notably, these suggestions are just examples to show that our theoretical results and contributions can offer valuable insights for further exploration, which can be used as theoretical guidance or support

for designs of the IQA models. Therefore, the theoretical results in the three proposed Theorems are the core contributions of this paper, rather than the three illustrative suggestions put forward.

## L  MORE EXPERIMENTS AND EXPERIMENTAL DETAILS

### L.1  MORE EXPERIMENTS

**Impacts of Different Hyper-parameters in Eq. (87)**    Based on our theoretical results, we introduce the Consistency Regularizer and the Parameter Regularizer in Eq. (87). For the Parameter Regularizer, we focus on the vector of weight parameters, since there are many layers in a deep BIQA network and constraining each would be impractical. We train a BIQA model with a ResNet-18 (He et al., 2016) backbone on KonIQ-10k (Hosu et al., 2020) and test on CID2013 (Virtanen et al., 2014). The PLCC and SRCC results for different choices of $\mu$ and $\nu$ are reported in Tables 1 and 2.

Table 1: The impacts of $\mu$ with fixed $\nu = 0.01$. BIQA model (ResNet-18 (He et al., 2016)) is trained on KonIQ-10k (Hosu et al., 2020) and tested on CID2013 (Virtanen et al., 2014).

| $\mu$ | 0 | 1 | 5 | 10 | 15 |
|------|------|------|------|------|------|
| PLCC | 0.681 | 0.685 | 0.702 | **0.713** | 0.708 |
| SRCC | 0.685 | 0.679 | 0.692 | **0.697** | 0.684 |
| RMSE | 13.37 | 14.16 | 13.05 | **12.21** | 12.74 |

Table 2: The impacts of $\nu$ with fixed $\mu = 10$. BIQA model (ResNet-18 (He et al., 2016)) is trained on KonIQ-10k (Hosu et al., 2020) and tested on CID2013 (Virtanen et al., 2014).

| $\nu$ | 0 | 0.01 | 0.05 | 0.1 | 1 |
|------|------|------|------|------|------|
| PLCC | 0.709 | **0.713** | 0.712 | 0.704 | 0.690 |
| SRCC | 0.687 | 0.697 | **0.701** | 0.695 | 0.678 |
| RMSE | 13.19 | 12.21 | **12.06** | 12.89 | 13.52 |

**Ablation Study for Suggestions in Appendix J**    In light of the theoretical results and related analysis, we offer three guidelines for designing BIQA models to demonstrate the practical value of our theorems. We denote $B$, $S_1$, $S_2$, and $S_3$ as the baseline ResNet-50 (He et al., 2016) (the same as in Figure 3 in Appendix L.3), Suggestion (**1**), Suggestion (**2**), and Suggestion (**3**), respectively, with $\mu = 10$ and $\nu = 0.01$. The results summarized in Tables 3 and 4 confirm the value of these suggestions and the soundness of our theoretical findings. The moderate effect of Suggestion (**3**) may stem from only considering weights in the prediction layer, while ignoring other network parameters.

Table 3: Ablation Study on KonIQ-10k (Hosu et al., 2020), it is divided into 8:2 for training and testing.

| Models | PLCC | SRCC | RMSE |
|--------|------|------|------|
| $B$ | 0.857 | 0.865 | 7.005 |
| $B+S_1$ | 0.863 | 0.872 | 6.846 |
| $B+S_1+S_2$ | 0.872 | 0.886 | 6.703 |
| $B+S_1+S_2+S_3$ | **0.873** | **0.892** | **6.674** |

Table 4:  Cross-data Ablation Study on KADID-10k (Virtanen et al., 2014) (train) and CID2013 (Virtanen et al., 2014) (test).

| Models | PLCC | SRCC | RMSE |
|--------|------|------|------|
| $B$ | 0.711 | 0.689 | 12.91 |
| $B+S_1$ | 0.718 | 0.705 | 12.27 |
| $B+S_1+S_2$ | **0.725** | 0.726 | **11.68** |
| $B+S_1+S_2+S_3$ | 0.719 | **0.731** | 11.80 |

**Experimental Verification of the Conclusion (1) from Theorem 3**    We train the BIQA network with backbone of ResNet-18 on KonIQ-10k, which is divided into two subsets with different distributions. Specifically, we divide the dataset into low-score set $LS$ and high-score set $HS$ based on the median of their MOS labels. Then $LS$ and $HS$ are then divided into 1:9 respectively, termed as $LS_1$, $LS_9$ and $HS_1$, $HS_9$. Subsequently, $LS_9$ and $HS_1$ are combined as $F_1$, $LS_1$ and $HS_9$ are combined as $F_2$, hence the distributions of $F_1$ and $F_2$ are different. In order to simulate various distribution differences between different training sets and test sets, we divided $F_1$ into 2 equal parts named $F_1^1$ and $F_1^2$ randomly, where $F_1^1$ is used as the training set, and $F_1^2$ and $F_2$ are combined as the test set. For the convenience of presentation, $F_1^{\text{test}}$:$F_2^{\text{test}}$ denotes the proportion of the two distributions $F_1$ and $F_2$ in the test set, and $F_1^{\text{train}}$ denotes the training set from distribution $F_1$. The whole construction process can be referred intuitively in Figure 2 in the Appendix L.2. According to Table 5, we can observe that the greater the distribution difference, the worse the generalization performance.

**Combination of Advanced BIQA Network with Design Suggestions**    To further verify the rationality of our theoretical results and suggestions about the generalization ability of BIQA models,

Table 5: The impact of distribution differences on generalization performances of BIQA model. $F_1^{\text{test}}:F_2^{\text{test}}$ denotes the proportion of the two different distributions in the test set. $F_1^{\text{train}}$ denotes the training set from $F_1$. The experiments are conducted on the dataset KonIQ-10k (Hosu et al., 2020).

| $F_1^{\text{train}}:(F_1^{\text{test}}:F_2^{\text{test}})$ | 1:(1:0) | 1:(1:0.5) | 1:(1:1) | 1:(1:2) | 1:(0:2) |
|---|---|---|---|---|---|
| PLCC | **0.818** | 0.775 | 0.746 | 0.723 | 0.695 |
| SRCC | **0.807** | 0.784 | 0.759 | 0.738 | 0.716 |
| RMSE | **8.308** | 8.621 | 9.248 | 9.874 | 10.122 |

Table 6: Results on KonIQ-10k (Hosu et al., 2020) dataset.

| Models | PLCC | SRCC |
|---|---|---|
| DBCNN (Zhang et al., 2020) | 0.884 | 0.875 |
| MetaIQA (Zhu et al., 2020) | 0.887 | 0.850 |
| MUSIQ (Ke et al., 2021) (our run) | 0.917 | 0.904 |
| MUSIQ (Ke et al., 2021)+$S_2$+$S_3$ | **0.922** | **0.910** |

Table 7: Results on SPAQ (Fang et al., 2020) dataset.

| Models | PLCC | SRCC |
|---|---|---|
| FRIQUEE (Ghadiyaram & Bovik, 2017) | 0.830 | 0.819 |
| DBCNN (Zhang et al., 2020) | 0.915 | 0.911 |
| MUSIQ (Ke et al., 2021) (our run) | 0.912 | 0.909 |
| MUSIQ (Ke et al., 2021)+$S_2$+$S_3$ | **0.914** | **0.916** |

we train the advanced BIQA network MUSIQ (Ke et al., 2021) with Suggestion **(2)** ($\mu = 10$) and Suggestion **(3)**[2] ($\nu = 0.005$) on SPAQ and KonIQ-10k, the results of which are compared with that of the original MUSIQ (Ke et al., 2021). We record experimental results on KonIQ-10k and SPAQ on Table 6 and Table 7 respectively. In addition, we compare our result with other baselines, including DBCNN (Zhang et al., 2020) and MetaIQA (Zhu et al., 2020) in the experiments on KonIQ-10k, and FRIQUEE (Ghadiyaram & Bovik, 2017) and DBCNN (Zhang et al., 2020) on SPAQ. The details of experimental settings are the same as described in MUSIQ (Ke et al., 2021). From Table 6 and Table 7, the results of the combination of MUSIQ (Ke et al., 2021) and Suggestions **(2)** and **(3)** are better than that of original MUSIQ (Ke et al., 2021), which confirms the soundness of our theoretical findings and suggestions.

### L.2 EXPERIMENTAL SETTINGS

**Implementation Details**  Our experiments are conducted with the Pytorch library on two Intel Xeon E5-2609 v4 CPUs and four NVIDIA RTX 2080Ti GPUs. The batch size $B$ is set as 64. The training is conducted for just 100 epochs in total with SGD optimization. Meanwhile, we resize all the images into $256 \times 256$ and randomly crop 10 sub-images to the size of $224 \times 224$. For the BIQA model with backbone of ResNet-18 in Table 1, Table 2 and Table 5, and BIQA model with backbone of ResNet-50 Table 3 and Table 4, we initialize the backbone by the pre-training weights obtained by classification task on ImageNet (Deng et al., 2009) before training. In the experiments of Table 3 and Table 4, we set $\mu = 10$ and $\nu = 0.01$. In cross-dataset experiments, there exist 5 epochs of fine-tuning before the test. In the experiments of Table 5, we do not apply our three suggestions to the BIQA model with backbone ResNet-18 since this part of the experiment aims to study the impact of changes in the distribution from the training set to the test set on generalization performance. Since most experiments in this paper are cross-data experiments, we normalized the MOS labels of all datasets to [1,100] before training and testing. The intuitive construction process of different distributions of KonIQ-10k for the verification of Theorem 3 are shown in Figure 2.

**Datasets**  In this paper, we perform experiments on five representative authentic image databases:

---

[2]Suggestion **(1)** is not considered here since the multi-scale features have already been incorporated in MUSIQ (Ke et al., 2021) during the training process.

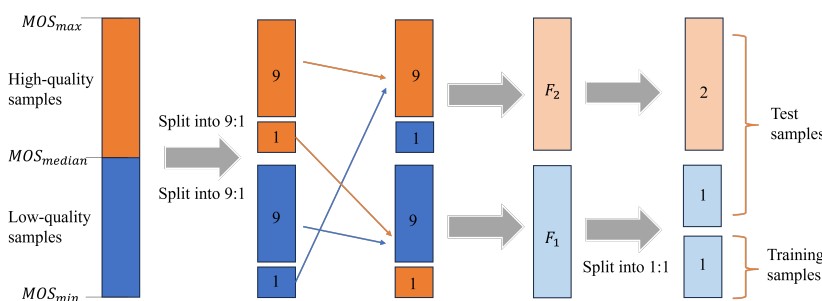

Figure 2: The intuitive construction process of different distributions of KonIQ-10k for the verification of Theorem 3 in experiments of Table 5 in the main text.

Table 8: Attributes of five typical IQA databases in experiments.

| Databases | Number | MOS Range | Distortion Type |
|---|---|---|---|
| TID2013 (Ponomarenko et al., 2015) | 3,000 | [0,9] | Synthetic |
| KADID-10k (Virtanen et al., 2014) | 10,125 | [1,5] | Synthetic |
| KonIQ-10k (Hosu et al., 2020) | 10,073 | [1,5] | Authentic |
| LIVE-C (Ghadiyaram & Bovik, 2015) | 1,162 | [0,100] | Authentic |
| CID2013 (Virtanen et al., 2014) | 480 | [0,100] | Authentic |
| SPAQ (Fang et al., 2020) | 11,125 | [0,100] | Authentic |

- KonIQ-10k (Hosu et al., 2020). It includes 10,073 images with authentic distortions chosen from YFCC100M (Thomee et al., 2016). Eight depth feature-based content or quality metrics are used in sampling process to ensure a wide and uniform distribution of image content and quality in terms of brightness, color, contrast and sharpness. And its quality is reported by MOS with the range of $[1, 5]$.

- LIVE-C (Ghadiyaram & Bovik, 2015). LIVE-C consists of 1,162 authentically distorted images captured from many diverse mobile devices. Each image was assessed on a continuous quality scale by an average of 175 unique subjects, and the MOS labels range in $[0, 100]$.

- TID2013 (Ponomarenko et al., 2015). This database contains 3,000 images, which are obtained from 25 reference images, 24 types of distortions for each reference image, and 5 levels for each type of distortion. The MOS labels range in $[0, 9]$

- SPAQ (Fang et al., 2020). SPAQ includes 11,125 images taken by 66 mobile phones, which contains a wide range of distortions during shooting, such as: sensor noise, blurring due to out-of-focus, motion blurring, over- or under-exposure, color shift, and contrast reduction. And the MOS labels range in $[0, 100]$.

- KADID-10k (Virtanen et al., 2014). It includes 81 pristine images, where each pristine image was degraded by 25 distortions in 5 levels. For each distorted image, 30 reliable degradation category ratings were obtained by crowdsourcing performed by 2,209 crowd workers. The MOS labels range in $[1, 5]$.

- CID2013 (Virtanen et al., 2014). CID2013 includes six image sets; on average, 30 subjects have evaluated 12–14 devices depicting eight different scenes for a total of 79 different cameras, 480 images, and 188 subjects. The MOS labels range in $[0, 100]$.

**Evaluation Metrics**    We evaluate BIQA models by two typical metrics, including Pearson Linear Correlation Coefficient (PLCC) and Spearman Rank-order Correlation Coefficient (SRCC). In addition, the $L_1$ loss is also used as one metric to study the impact of the different learning level of quality perception features.

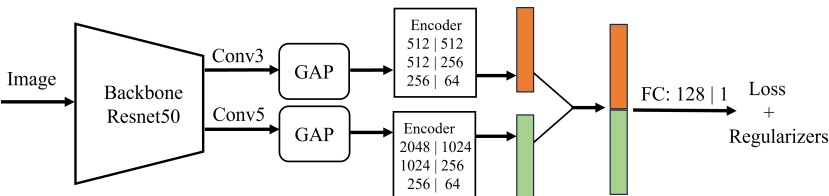

Figure 3: The design example that fuses multi-level features for BIQA model under suggestion (1). $C_1|C_2$ denotes a fully-connected (FC) layer mapping from dimension $C_1$ to $C_2$.

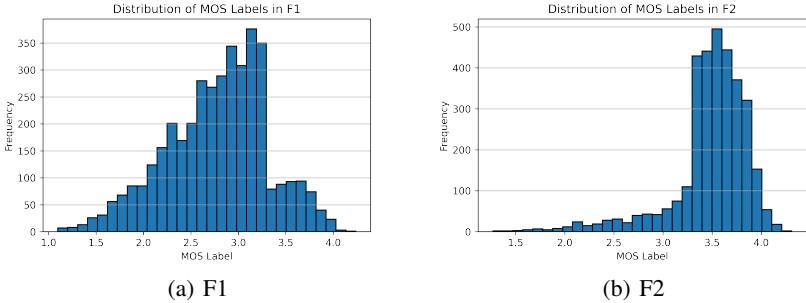

(a) F1                   (b) F2

Figure 4: Distributions of MOS values in F1 (a) and F2 (b) on KonIQ-10k.

### L.3    DESIGN EXAMPLE AND OTHER RESULTS

**An Example of Our Suggestions for BIQA Model Design**    Figure 3 is an example with Resnet-50 (He et al., 2016) as the backbone, which fuses 2 levels of features.

**Distribution of MOS values of F1 and F2 from KonIQ-10k**    In the Experimental Verification of Theorem 3, to study the impact of changes in the distribution from the training set into the test set on generalization performance, we have divided KonIQ-10k (Hosu et al., 2020) to two subsets with different distributions named F1 and F2. Figure 4 shows the distributions of MOS values in F1 and F2 on KonIQ-10k (Hosu et al., 2020).

### L.4    MORE DETAILED DESCRIPTION ABOUT EQ. (87)

In Eq. (87), $x_f$ denotes the extracted quality perception feature vector, and $R_B(x_f)$ represents the order (from smallest to largest) of the Euclidean distances between $x_f$ and the feature vectors of other image samples in the same batch $B$. $y$ denotes the ground truth MOS label for $x_f$, and $R_B(y)$ represents the order (from smallest to largest) of the absolute distances between $y$ and the MOS labels of other image samples in the same batch $B$. The absolute distances are computed by the absolute difference between two MOS scalars. $\cos$ refers to the cosine similarity. $Norm$ denotes the max-min normalization, for an original random variable $x$, that is:

$$Norm(x) = \frac{x - x_{\min}}{x_{\max} - x_{\min}} \tag{88}$$

where the maximum value of the cosine similarity in Eq. (87) is 1, and the minimum value f the cosine similarity in Eq. (87) is $\cos([1, 2, \ldots, B-1], [B-1, B-2, \ldots, 1])$.

**The second term in Eq. (87) means the consistency regularizer, corresponding to Suggestion (2)**, which directly minimizes empirical error by maintaining the consistency between the feature space and label space. The core idea stems from the conflict between strong representation and generalization in BIQA models revealed by Theorem 1 and Theorem 3. To avoid this conflict, instead of enhancing the model's representation capacity by increasing network depth, we focus on directly minimizing empirical error while maintaining good generalization. Therefore, the proposed consistency regularizer is an intuitive and effective choice. **The third term in Eq. (87) means the parameter regularizer, corresponding to Suggestion (3)**, where $\mathbf{W}_L$ denotes the weight parameter of the final prediction layer, and $|\cdot|_F$ is the Frobenius norm. $\mu$ and $\nu$ are the hyper-parameters.

Table 9: The MAE performances of our suggestions in this paper on the UTK-Face dataset (Zhang et al., 2017) in the age prediction task.

| Models | $B$ | $B + S_1$ | $B + S_1 + S_2$ | $B + S_1 + S_2 + S_3$ |
|--------|-----|-----------|------------------|------------------------|
| MAE | 4.96 | 5.03 | 4.88 | 4.91 |

According to Appendix J, the proposed theorems can not only provide theoretical support for existing works, but also offer valuable insights for further exploration. Therefore, what we wish to emphasize is that: the proposed loss function in Eq. (87) just serves as the practical examples for the guidance of BIQA network design based on these theorems. In our future work, we will explore more comprehensive generalization theories for BIQA models and uncover practical values and insights to guide the design of future deep learning-based BIQA models.

## M  APPLICABILITY OF OUR SUGGESTIONS TO OTHER REGRESSION TASKS.

Although the network settings and loss functions may not be IQA-specific, the theoretical analysis and contributions in this paper have fully considered the task characteristics of the IQA domain. This is one of the core differences of this paper, distinguishing it from existing theoretical research, and it mainly includes the following two aspects.

- Focus on Regression Tasks in IQA: As stated in Appendix B.2, most existing theoretical studies on deep neural networks focus on fully connected networks. Although some recent works have explored the generalization of CNNs, they are primarily applicable to classification tasks rather than regression tasks. However, the IQA task studied in this paper is a classic regression problem, making prior theoretical results for classification tasks unsuitable for the IQA tasks.

- Consideration of IQA-Specific Characteristics: As discussed in Section 5, this paper thoroughly considers the unique characteristics of IQA tasks: (1) quality perception information predominantly resides in low-level image features, and (2) effective representation learning of multi-level image features and distortion information is critical for the generalization of Blind IQA (BIQA) methods. In contrast, existing theoretical studies on general deep neural networks often overlook the role of low-level image features.

To illustrate this more intuitively, we conducted an experiment on another regression task, i.e. on the UTK-Face dataset (Zhang et al., 2017), where the task is to predict age from input facial images. Using ResNet-50 as the backbone, 80% of the data was used for training, and 20% for testing. The experimental setup followed Table 3, and the recorded MAE (Mean Absolute Error) results for $B$, $B + S_1$, $B + S_1 + S_2$, $B + S_1 + S_2 + S_3$ as follows:

From Table 9, we can observe that the three suggestions proposed in this paper perform poorly in the age prediction task. This is probably because the age prediction task primarily relies on high-level features about the age of the face in the input image. This indirectly confirms that the contributions of this paper are more relevant to IQA tasks, which differ significantly from other tasks.

