# OpenReview forum: "What Governs the Quality-Aware Generalization and Representation Capacity in No-Reference Image Quality Assessment Models?"
_ICLR.cc/2026/Conference — ICLR 2026 Conference Withdrawn Submission_

### Official Review · Reviewer_iDio · 2025-10-28

**Soundness:** 4
**Presentation:** 3
**Contribution:** 3
**Rating:** 6
**Confidence:** 4

**Summary:**

Despite success with deep learning, BIQA models often fail to generalize due to limited labeled data and lack of theoretical understanding of what drives generalization and feature representation. The paper provides a theoretical framework to analyze what governs generalization and representation capacity in BIQA models. It establishes four theorems, leading to the following findings: Deeper networks or larger kernels increase generalization error, Low-level features are crucial for generalization, Larger distribution shifts degrade generalization, Low-level features with lower intrinsic dimensions improve robustness, High-level features and deeper networks enhance representation capacity.

**Strengths:**

+ This work provides one of the first formal theoretical frameworks analyzing generalization and representation capacity in BIQA models — an area previously dominated by empirical work.
+ It reveals a fundamental conflict between generalization (favored by low-level features) and representation capacity (favored by high-level features).
+ It offers actionable design guidelines for BIQA development.

**Weaknesses:**

- The theoretical results are derived on a simplified CNN-based architecture, ignoring modern architectures like ViT.
- While the paper motivates the work by citing data scarcity, it does not propose solutions (e.g., semi-supervised learning, synthetic data) to address it directly.
- The heavy mathematical exposition may limit accessibility for practitioners focused on applied computer vision or engineering.

**Questions:**

N.A.

---

### Official Review · Reviewer_jW5o · 2025-10-31

**Soundness:** 2
**Presentation:** 1
**Contribution:** 2
**Rating:** 2
**Confidence:** 4

**Summary:**

The paper develops a theoretical framework for blind image quality assessment (BIQA) models, establishing generalization bounds and a representation-capacity bound. Using covering numbers and VC dimension (Theorems 1–2), it shows deeper networks and larger kernels loosen generalization. Under distribution shift, a tighter bound via Rademacher complexity and intrinsic dimension (Theorem 3) quantifies how train-test divergence degrades performance. Conversely, a PAC-Bayes analysis (Theorem 4) links higher-level features and model size to greater representation capacity (lower empirical error), revealing a fundamental tension: features that improve representation can harm generalization. Experiments across IQA datasets empirically validate these predictions and suggest training strategies that balance the trade-off.

**Strengths:**

1. The paper develops fine- and coarse-grained generalization bounds and a PAC Bayes representation capacity bound, showing that low-level features promote generalization while higher-level features strengthen representation, and exposing a fundamental tension between the two

2. This paper provides detailed proofs for each proposed theorem.

**Weaknesses:**

1. All proofs are pretty standard in statistical machine learning, with few theoretical insights.

2. The assumptions made for proofs may not hold in the real world, giving rise to limited practical relevance.

3. Following Comment 2, although the paper proposes theorems on quality-aware generalization and representation, the experimental evaluation is limited and insufficient to demonstrate their effectiveness. Moreover, restricting the comparison to CNN-based methods fails to represent the current landscape of DL-based NR-IQA, which is increasingly driven by Transformer-based and hybrid (e.g., VLM)  approaches.

4. The paper’s structure makes it difficult to discern the main contributions and insights.

5. Tables 3, 4, 6, and 7 show only minimal gains over the baseline, which weakens support for the proposed theorem.

**Questions:**

1. The bound in $\textbf{Theorem 3}$ depends on the $\chi^2$ divergence $D(P_{\text{test}} \| P_{\text{train}})$ and the “intrinsic dimension” of low-level features $h(\Sigma) = \mathrm{tr}(\Sigma)/\|\Sigma\|_2$ (Eq.(12), p.7; Appendix H, 22--24). However, the paper \textbf{does not} estimate these two quantities in the experiments: Table 5 (p. 28) only constructs different distribution ratios of KonIQ-10k, and then qualitatively shows that “a larger ratio difference $\rightarrow$ worse performance,” without reporting any density ratio/divergence estimates or PCA estimates of $h(\Sigma)$. It also does not clarify from which specific layer the “low-level features” are taken, nor how consistency with the theoretical $\Sigma$ is ensured. On the other hand, the bound also involves $M = \prod_j \|W_j\|_F$ (Appendix H, Eq.(53)), while the author’s “Recommendation (3)” only penalizes the last layer’s weights (Eq.(87), p. 27), which is inconsistent with the theoretical control over the product of all layers.

2. In Appendix I, the proof of Theorem 4 assumes that constraining the output scores to $[0, r]$ implies a per-parameter posterior-mean bound $|\mu_{(l,i)}| \le r$ (Eq.(84), p.25), which is then used to upper-bound the KL divergence and conclude $\mathrm{KL}(Q \|\| P) \le O(k (\sum_l m_l) r^2)$ (Eq.(86), p.25). However, an output-range constraint does not by itself imply bounds on each weight-mean component; a link like layerwise Lipschitz/spectral-norm control would be required to carry output bounds back to parameter means. Moreover, the experiments use deterministic SGD (Appendix L.2: SGD optimization, p. 28) and never define or estimate a posterior $Q$ (e.g., SWA/SWAG or a Laplace approximation), so the claimed KL-capacity story is not empirically grounded.

---

### Official Review · Reviewer_VGAE · 2025-11-01

**Soundness:** 2
**Presentation:** 2
**Contribution:** 2
**Rating:** 2
**Confidence:** 5

**Summary:**

The paper investigates the theoretical foundations of no-reference image quality assessment (NR-IQA) by analyzing the generalization and representation capacities of deep neural networks through established learning theory tools. It presents several theorems based on covering numbers, VC dimension, Rademacher complexity, and PAC-Bayes analysis to formalize how network depth and feature hierarchy influence performance. The authors conduct experiments on standard IQA datasets to validate the theoretical observations and offer design recommendations for improving model generalization.

**Strengths:**

1.	This paper studies an underexplored but relevant topic, and tries to provide a theoretical perspective on generalization in NR-IQA models.

2.	Experiments are conducted on multiple standard IQA datasets, and results are consistent with the general trends predicted by the theoretical observations.

**Weaknesses:**

1.	The paper mainly reuses well-established theoretical frameworks (covering number, VC dimension, Rademacher complexity, PAC-Bayes) to revisit known principles in IQA. It essentially provides an interpretation of existing empirical consensus rather than introducing new theoretical insights or methodological advances. Moreover, the proposed analysis remains largely generic and shows limited relevance to IQA-specific characteristics.

2.	Multiple theorems largely restate similar ideas about the generalization–representation trade-off using different frameworks. This redundancy inflates complexity without deepening understanding. Consolidating the tightest bounds would make the argument more focused and meaningful.

3.	The analysis attributes generalization differences to “multi-level feature fusion,” but the results fundamentally reflect how model capacity varies with depth. Comparing networks of different depths cannot support claims about hierarchical features within the same model.

4.	Several issues weaken the validity of the theoretical results:

Theorem 1: $\frac{1}{n} \sqrt{\sum^L_{l=1} k_l d_l log⁡(a/\epsilon)+log⁡(1/\delta)}$ should be $\sqrt{ \frac{\sum^L_{l=1} k_l d_l log(a/\epsilon)+log⁡(1/\delta)}{n} }$.

Theorem 4: Relies on the PAC-Bayes framework with Gaussian Prior and Posterior for Bayesian CNNs, while experiments are conducted with CNN-based models. This mismatch makes the theorem inapplicable to the reported results.

5.	The reported performance gains are minimal (less than 0.01 in Table 6 and Table 7), offering little practical significance.

**Questions:**

Refer to Weaknesses.

---

### Official Review · Reviewer_vcum · 2025-11-03

**Soundness:** 3
**Presentation:** 2
**Contribution:** 2
**Rating:** 4
**Confidence:** 3

**Summary:**

This paper aims to provide a theoretical framework for understanding the generalization and representation capacity of deep learning–based Blind Image Quality Assessment (BIQA) models. The authors argue that while empirical BIQA methods have improved significantly, their theoretical foundations—especially regarding why certain architectures generalize better—remain unclear.
To address this, they derive several upper bounds. The authors conclude that BIQA models face an intrinsic trade-off between generalization and representation—low-level features favor generalization, while high-level ones favor expressiveness.
They further validate these theorems through experiments on standard datasets (KonIQ-10k, LIVE-C, TID2013) and discuss implications for BIQA network design.

While the paper is well-written and mathematically correct, it lacks originality and theoretical depth. The analyses are derivative of established machine-learning theory and do not provide new understanding of BIQA mechanisms or meaningful practical guidance for model design. The claimed “first theoretical treatment” of BIQA generalization may be true in a narrow sense, but the contribution is incremental and self-contained, with limited impact beyond formal restatement.

**Strengths:**

[1] Comprehensive Theoretical Ambition. The paper attempts to connect multiple theoretical lenses (Covering Number, VC Dimension, Rademacher Complexity, PAC-Bayes) within the BIQA context—an area rarely formalized mathematically. This unified treatment is intellectually ambitious and potentially relevant for the IQA research community.

[2] Clear Problem Motivation. The paper correctly identifies a real gap in the literature—most IQA works focus on empirical design without theoretical explanation of performance differences.

[3] Structured Presentation. The theorems and assumptions are carefully defined with consistent mathematical notation, providing a relatively clear reading experience for a mathematically inclined audience.

[4] Empirical Corroboration. The experimental plots do reflect the expected qualitative trends (e.g., decreasing empirical error but increasing test error with depth), lending some credibility to the theoretical statements.

**Weaknesses:**

[1] Lack of True Theoretical Innovation. Despite the mathematical formalism, each theorem is a direct adaptation of classical results from statistical learning theory to the BIQA setting, with only superficial contextualization. Theorems 1–3 are largely textbook applications of Covering Number, VC, and Rademacher complexity, with IQA-specific notation inserted. The PAC-Bayes analysis (Theorem 4) follows standard derivations for Gaussian posteriors without novel adaptation to the peculiarities of image quality perception or regressive loss functions. Thus, the “theoretical contribution” is more a restatement than an advancement of learning theory.

[2] Weak Practical Impact and Interpretability. The resulting theorems—e.g., “generalization decreases with model depth”—are unsurprising and already well established. The paper fails to translate these bounds into actionable guidance beyond generic statements like “low-level features help generalization.”

[3] Over-simplified Model Assumptions. The analyses assume an “unembellished CNN” with bounded weights and 1-Lipschitz activations, which abstracts away essential BIQA design realities such as patch-based learning, attention modules, transformers, or multi-scale fusion. Consequently, the derived bounds may not generalize to state-of-the-art architectures (MUSIQ, Hyper-IQA, Re-IQA, etc.).

[4] Questionable Empirical Validation. The experiments are rudimentary—small-scale CNNs and basic cross-dataset testing. There is no quantitative fit between theoretical constants and empirical results, so “verification” remains qualitative at best.

**Questions:**

I have list my concerns in the Weakness.

---

### Note · Authors · 2025-12-01

I have read and agree with the venue's withdrawal policy on behalf of myself and my co-authors.